**EMBO** *reports*

# Cep120 is essential for kidney stromal progenitor cell growth and differentiation

Ewa Langner [1], Tao Cheng[1], Eirini Kefaloyianni [2], Charles Gluck[1], Baolin Wang [3] & Moe R Mahjoub [1,4✉]

## Abstract

**Mutations in genes that disrupt centrosome structure or function can cause congenital kidney developmental defects and lead to fibrocystic pathologies. Yet, it is unclear how defective centrosome biogenesis impacts renal progenitor cell physiology. Here, we examined the consequences of impaired centrosome duplication on kidney stromal progenitor cell growth, differentiation, and fate. Conditional deletion of the ciliopathy gene Cep120, which is essential for centrosome duplication, in the stromal mesenchyme resulted in reduced abundance of interstitial lineages including pericytes, fibroblasts and mesangial cells. These phenotypes were caused by a combination of delayed mitosis, activation of the mitotic surveillance pathway leading to apoptosis, and changes in both Wnt and Hedgehog signaling that are key for differentiation of stromal cells. Cep120 ablation resulted in small hypoplastic kidneys with medullary atrophy and delayed nephron maturation. Finally, Cep120 and centrosome loss in the interstitium sensitized kidneys of adult mice, causing rapid fibrosis after renal injury via enhanced TGF-β/Smad3-Gli2 signaling. Our study defines the cellular and developmental defects caused by loss of Cep120 and aberrant centrosome biogenesis in the embryonic kidney stroma.**

**Keywords** Centriole; Centrosome; Fibrosis; Interstitium; Stromal Mesenchyme
**Subject Categories** Cell Cycle; Development; Molecular Biology of Disease

## Introduction

The centrosome is the main microtubule-organizing center in mammalian cells, essential for regulating the interphase and mitotic microtubule arrays, and is indispensable for cilium formation (Breslow and Holland, 2019). Cilia are also microtubule-based organelles that are templated by the centrosome, and perform important chemo- and mechanosensory functions in cells (Delling et al, 2016; Liu et al, 2023; Shah et al,

2009). Under normal physiological conditions, there is typically one centrosome-cilium complex per cell, and this number is tightly controlled (Breslow and Holland, 2019; Loncarek and Bettencourt-Dias, 2018; Nigg and Holland, 2018; Nigg and Stearns, 2011). The process of centrosome biogenesis is strictly regulated throughout each cell cycle, with centrosome duplication initiating in S-phase and centrosome segregation in mitosis (Breslow and Holland, 2019; Loncarek and Bettencourt-Dias, 2018). Mutations in genes that disrupt the centrosome duplication cycle can result in daughter cells that contain aberrant centrosome number, and lead to pathological phenotypes (Conduit et al, 2015; Jaiswal and Singh, 2021; Nigg and Holland, 2018). Specifically, mutations in centriole duplication factors can block centrosome biogenesis and result in cells lacking centrosomes after several rounds of cell division, a phenomenon termed **C**entrosome **L**oss (CL) (Fong et al, 2016; Lambrus et al, 2016; Meitinger et al, 2016; Mikule et al, 2007; Wong et al, 2015).

Several cellular and molecular changes are known to occur following CL. Since centrosomes facilitate the assembly of the mitotic spindle, cell cycle progression and mitosis are often impaired in cells lacking centrosomes (Conduit et al, 2015; Loncarek and Bettencourt-Dias, 2018). Interestingly, this effect may be cell type, tissue, or organ specific. CL in normal cells leads to prolonged mitosis, p53-dependent cell cycle arrest and activation of the p53BP1-USP28-TP53-dependent mitotic surveillance mechanism, leading to caspase-mediated apoptosis (Fong et al, 2016; Lambrus et al, 2016; Mikule et al, 2007; Wong et al, 2015). In vivo, inducing CL globally in mice causes prometaphase delay and p53-dependent apoptosis in the majority of cells in the embryo, which prevents its development upon midgestation, and causes lethality by embryonic day nine (Bazzi and Anderson, 2014). Conditional induction of CL in neural progenitor cells during brain development similarly results in delayed mitosis and activation of the mitotic surveillance pathway (Phan et al, 2021). This causes an increased frequency of progenitor cell death, and their premature differentiation into neurons, thus depleting the neural progenitor pool and ultimately leading to small brain (microcephaly) phenotypes (Phan et al, 2021). In the embryonic lung, inducing CL in the developing endoderm similarly causes p53-mediated apoptosis, but only in proximal airway cells with low extracellular signal-regulated kinase (ERK) activity (Xie et al, 2021).

---

[1]Department of Medicine (Nephrology Division), Washington University, St Louis, MO, USA. [2]Department of Medicine (Rheumatology Division), Washington University, St Louis, MO, USA. [3]Department of Genetic Medicine, Weill Medical College of Cornell University, New York, NY, USA. [4]Department of Cell Biology and Physiology, Washington University, St Louis, MO, USA. ✉E-mail: mmahjoub@wustl.edu

Furthermore, centrosomes of the endoderm appear to be dispensable for progenitor cell growth and differentiation during development of the intestines (Xie et al, 2021). Thus, the consequences of CL differ depending on the specific type of the cell, tissue, and developmental context.

One gene that is essential for centrosome biogenesis and duplication is Cep120. It encodes a daughter centriolar-enriched protein that, when depleted, leads to defective centriole (and thus centrosome) duplication, and causes CL following cell division (Comartin et al, 2013; Lin et al, 2013; Mahjoub et al, 2010; Tsai et al, 2019). In contrast, Cep120 depletion in non-dividing quiescent cells disrupts centrosome homeostasis, causing excessive pericentriolar material accumulation, defective microtubule nucleation and dynein-dependent cargo trafficking, ultimately leading to aberrant ciliary assembly and signaling (Betleja et al, 2018; Joseph et al, 2018). In vivo, genetic ablation of Cep120 globally results in early embryonic lethality in mice (Wu et al, 2014). Conditional deletion of Cep120 in the central nervous system causes hydrocephalus and cerebellar hypoplasia (Wu et al, 2014), while siRNA-mediated depletion of Cep120 in the developing mouse brain disrupts growth and self-renewal of the neural progenitor pool during neocortical development (Xie et al, 2007). These phenotypes are due to failed centrosome duplication, aberrant microtubule nucleation and organization, abnormal maturation of cerebellar granule neuron progenitors (CGNPs), and defective ciliogenesis in differentiated ependymal cells (Wu et al, 2014; Xie et al, 2007). Moreover, Cep120 mutations were recently identified in two ciliopathy syndromes, namely Joubert Syndrome (JS) and Jeune Asphyxiating Thoracic Dystrophy (JATD); patients display multi-organ pathologies including severe congenital renal developmental defects and fibrotic kidney disease (Roosing et al, 2016; Shaheen et al, 2015). Yet, how the mutations in Cep120 cause the kidney development and fibrocystic disease phenotypes in humans remains unclear.

Although the consequences of defective centrosome biogenesis in some organs have been determined, the outcomes of mutations that cause CL during kidney development are not known. Embryonic kidney formation relies on reciprocal signaling between three distinct progenitor cell types: the cap mesenchyme (CM) which contains nephron progenitor cells (NPC) that give rise to the proximal segments of mature nephrons, the ureteric bud (UB) epithelia which gives rise to the collecting duct segments, and stromal mesenchyme (SM) progenitor cells that form the various interstitial lineages of the kidney (Little and McMahon, 2012; McMahon, 2016). Defects in the growth, differentiation, or function of each progenitor cell type disrupts the overall development of the kidney (Jain and Chen, 2019; McMahon, 2016). The SM differentiate into several stromal/interstitial cell lineages of the mature kidney, including interstitial pericytes and fibroblasts, mesangial cells and vascular smooth muscle cells (Li et al, 2014; Rowan et al, 2017). However, it is unclear whether the observed renal dysplasia and fibrotic scarring that occurs in JS and JATD patients is a consequence of centrosome gene mutations that impact the stromal mesenchyme, NPC or UB progenitors. Similarly, it is not known whether the onset of cystogenesis is due to mutations in centrosome genes in the nephron epithelia, or potentially induced by defective paracrine signaling from the stromal compartment upon centrosome loss.

Here, we investigated the consequences of CL in the stromal progenitor niche of the developing kidney. We found that Cep120 deletion in embryonic kidney stromal progenitors disrupts centrosome biogenesis in derived cell types including pericytes, interstitial fibroblasts, mesangial, and vascular smooth muscle cells. CL resulted in reduced abundance of several stromal cell populations (interstitial pericytes, fibroblasts and mesangial cells), leading to development of smaller kidneys with visible signs of medullary atrophy and delayed nephron maturation by postnatal day 15. The reduced interstitial cell populations were due to a combination of defective cell cycle progression of SM lacking centrosomes, p53-mediated apoptosis, and changes in signaling pathways essential for differentiation of stromal lineages. There was no spontaneous fibrosis or ECM deposition, however we observed nephron tubule dilations in kidneys of Cep120-KO mice at 5 months of age. Finally, we found that CL in the interstitium sensitized kidneys of adult mice, causing rapid fibrosis via enhanced TGF-β/Smad3-Gli2 signaling after renal injury.

## Results

### Cep120 deletion in embryonic kidney stromal progenitors disrupts centrosome biogenesis in derived cell types

To conditionally block centrosome duplication in stromal progenitor cells (Fig. 1A) of the embryonic kidney, we used a recently developed mouse model harboring a floxed allele of Cep120 (Wu et al, 2014; Fig. 1B). We crossed Cep120$^{F/F}$ mice to a FoxD1-Cre strain (hereafter referred to as Cep120-KO) that expresses Cre-recombinase in the kidney stromal mesenchyme (SM) progenitor cells beginning at E11.5 (Hatini et al, 1996; Humphreys et al, 2010). In wild-type kidneys of E15.5 mice, Cep120 staining was evident at the centrosome in Aldh1a2-positive stromal mesenchyme (Fig. 1C). Cep120 deletion in FoxD1-progenitors resulted in 52% decrease in the fraction of cells expressing Cep120 (Fig. 1C,D), and a concurrent loss of centrosomes as marked with γ-tubulin (Fig. 1E,F) or with Ninein/Centrin (Fig. 1G,H). The loss of Cep120 signal persisted postnatally in cell types derived from the FoxD1+ stromal progenitors; 52% of interstitial pericytes and fibroblasts (Figure EV1A,B), 49% of glomerular mesangial cells (Figure EV1D,E), and 64% of vascular smooth muscle cells (Fig. EV1G). Concomitantly, there was loss of centrosomes evident in the same cell populations; 40% of interstitial pericytes and fibroblasts (Figure EV1A,C), 55% of glomerular mesangial cells (Figure EV1F), and 52% of vascular smooth muscle cells (Figure EV1H). Importantly, Cep120 expression and centrosome assembly were unaffected in cells not derived from FoxD1 progenitors (e.g., tubular epithelia and endothelial cells; Figure EV1I,J). Thus, ablation of Cep120 in the stromal progenitor population causes loss of centrosomes specifically in FoxD1+ progenitors and their lineages.

### Loss of Cep120 and centrosomes results in small kidneys and decreased FoxD1-derived stromal cells populations

Next, we examined the consequences of centrosome loss in stromal cells on overall kidney development and function. Cep120-KO mice were born in the expected Mendelian ratios (Table EV1). At birth, Cep120-KO pups were similar in size to control littermates, however the mice were runted and significantly smaller by P15

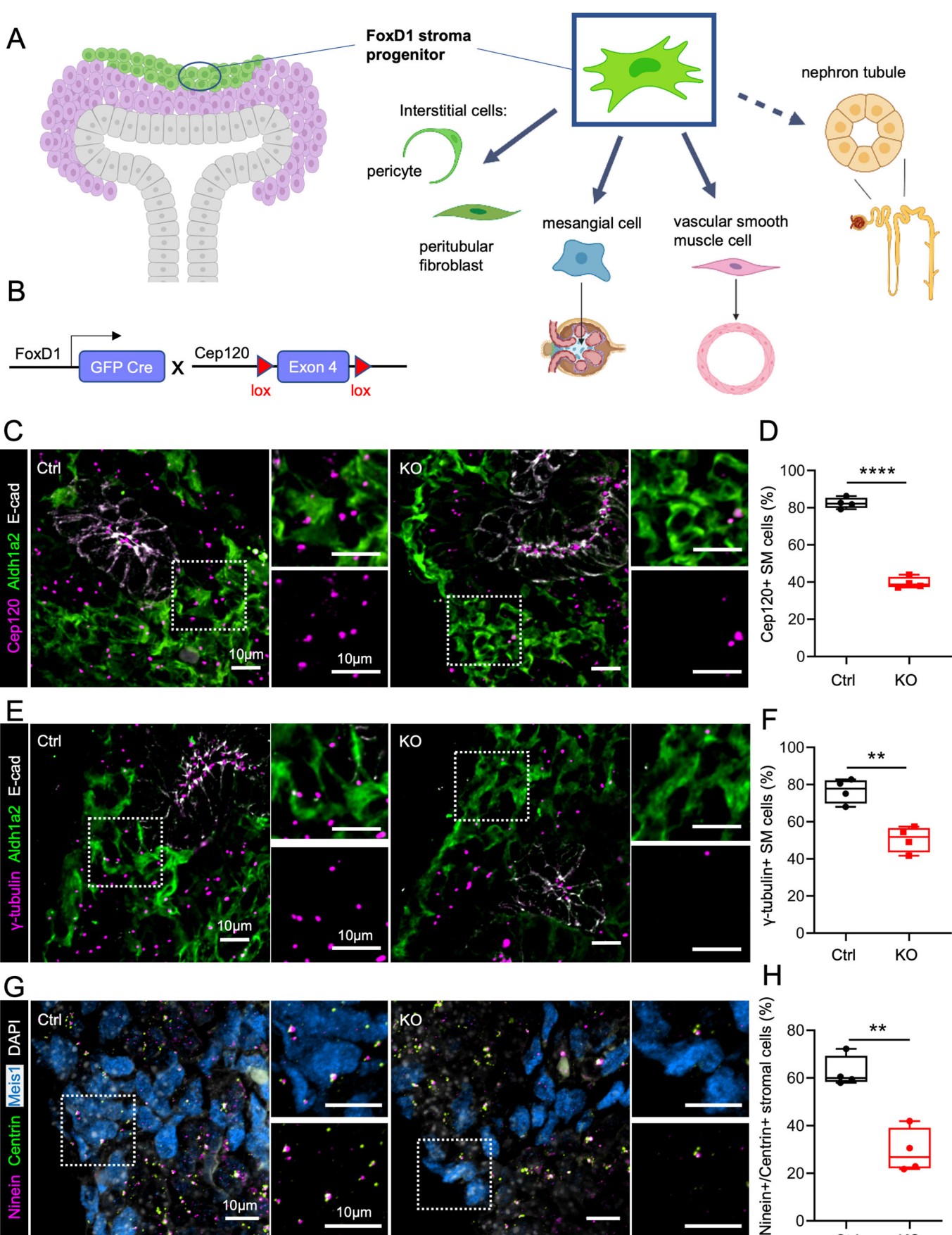

**Figure 1.  Cep120 deletion disrupts centrosome biogenesis in embryonic kidney stromal progenitors.**

(A) Schematic showing kidney stromal cell populations that are derived from FoxD1-positive progenitors. (B) Schematic of Cep120 conditional deletion in the stromal mesenchyme using FoxD1-Cre. (C) Immunofluorescence staining of E15.5 embryonic kidney sections with antibodies to mark Cep120, stromal mesenchyme (SM, Aldh1a2) and ureteric bud epithelium (E-cadherin). (D) Quantification of Aldh1a2-positive SM cells containing Cep120 at E15.5. $N = 878$ cells (Ctrl) and $N = 830$ (Cep120-KO). (E) Immunofluorescence staining of E15.5 embryonic kidney sections with antibodies to mark centrosomes ($\gamma$-tubulin), SM (Aldh1a2) and ureteric bud epithelium (E-cadherin). (F) Quantification of Aldh1a2-positive SM cells with centrosomes at E15.5. $N = 804$ cells (Ctrl) and $N = 733$ (Cep120-KO). (G) Immunofluorescence staining of E15.5 embryonic kidney sections with antibodies to mark centrioles (Centrin), centrosomes (Ninein) and stromal cells (Meis1). (H) Quantification of Meis1-positive stromal cells with centrosomes at E15.5. $N = 917$ cells (Ctrl) and $N = 920$ (Cep120-KO). Data information: $N = 4$ mice per group. A two-tailed unpaired $t$ test was used for analyses and $p$ value denoted as follows: $**p < 0.01$, $****p < 0.0001$. The vertical segments in the box plots show the first quartile, median, and third quartile. The whiskers on both ends represents the maximum and minimum for each dataset analyzed. Source data are available online for this figure.

(Fig. 2A). Analysis of survival rates indicated that some Cep120-KO mice died shortly after P15 (we refer to these as short-term survivors), while others lasted up to 5 months (long-term survivors; Fig. 2B). The short-term survivors typically exhibited strong extra-renal phenotypes (e.g. dome-shaped head, short snout, malocclusions), which were less pronounced in the long-term survivors. This has been previously reported when ablating genes using FoxD1-Cre-expressing mice, since FoxD1 is also expressed in other organs (Karolak et al, 2018; Nie and Arend, 2017). Interestingly, kidneys isolated from P15 Cep120-KO mice were smaller in size compared to controls (Fig. 2C,E). Moreover, we observed dilations in collecting duct tubules and signs of medullary atrophy (Fig. 2C,D). However, there was no significant decline in kidney function. Quantification of blood urea nitrogen (BUN) and serum creatinine levels showed a slight increase, but within physiological range (<42 mg/dl and <0.25 mg/dl, respectively) (Fig. 2F,G). Similarly, there was no proteinuria observed in Cep120-KO mice as determined by albumin to creatinine ratio in the urine (Figs. 2H and EV2A,B).

Next, we sought to determine the cause of the reduced kidney size and the effects of centrosome loss on the stromal progenitor-derived populations of the adult kidney. Kidneys of P15 mice were stained with markers of pericytes and interstitial fibroblasts (marked by PDGFR-$\beta$, desmin and $\alpha$-SMA; (Smith et al, 2012), and glomerular mesangial cells (marked by desmin and GATA3; (Grigorieva et al, 2019; Vaughan and Quaggin, 2008). There was a significant decrease in the abundance of tubulointerstitial pericytes and fibroblasts in the cortical region of Cep120-KO kidneys (28% decrease in PDGFR-$\beta$ and desmin double-positive pericytes and 57% loss in $\alpha$-SMA-positive fibroblasts) (Fig. 3A–C). In addition, there was 59% decrease in $\alpha$-SMA-positive pericytes surrounding the Bowman's capsule of glomeruli (Figure EV2C,D), further indication of abnormal pericyte specification. Similarly, quantification of mesangial cell abundance in glomeruli showed a significant (23%) decrease upon centrosome loss (Fig. 3D–F). Moreover, we noted upregulated expression of $\alpha$-SMA in the mesangium of Cep120-KO kidneys at P15 (Figure EV2C,E), suggesting a delay in mesangial cell maturation (Vaughan and Quaggin, 2008). Together, these data indicate that loss of centrosomes in stromal progenitors leads to a decrease in the formation of their derived cell populations, resulting in smaller kidney size and defective medullary morphology.

## Cep120 and centrosome loss causes a $G_2$/M delay and promotes apoptosis of stromal progenitors

To determine how the loss of stromal cells observed in postnatal Cep120-KO kidneys occur, we measured the abundance and cell cycle profile of FoxD1+ progenitors. Immunofluorescence staining of E15.5 Cep120-KO kidneys showed a 30% decrease in Aldh1a2-positive SM cells (Fig. 4A,C), suggesting potential defects in their proliferation. To test this, we injected pregnant females with EdU which labels replicating DNA in cells undergoing S-phase (Fig. 4B; Pereira et al, 2017). The samples were co-stained with antibodies against phospho-histone H3 (pHH3), a marker of $G_2$/M-phases of the cell cycle (Hendzel et al, 1997; Ren et al, 2018). There was a marked increase in the fraction of $G_2$/M (Aldh1a2+/pHH3+) SM cells in the Cep120-KO kidneys (Fig. 4D). In contrast, there was no significant difference in the fraction of cells in $G_0$/$G_1$ (Aldh1a2+/EdU−/pHH3−) or S-phase (Aldh1a2+/EdU+) between groups (Fig. 4E,F). These data suggest that centrosome loss in stromal progenitors results in delayed transition through $G_2$/M, consistent with previous reports showing mitotic delay upon defective centrosome biogenesis (Chang et al, 2021; Fong et al, 2016; Phan et al, 2021).

It has recently been shown that centrosome loss and prolonged mitosis leads to p53-dependent cell cycle arrest, induction of the mitotic surveillance mechanism, and activation of caspase-dependent cell death (Chang et al, 2021; Fong et al, 2016; Phan et al, 2021; Poulton et al, 2019; Xie et al, 2021). We wondered if the observed $G_2$/M-phase delay in Cep120-KO kidneys similarly results in p53 activation and increased apoptosis of stromal progenitors and derived cells. Co-staining stromal cells of E15.5 Cep120-KO kidneys with p53 showed a 6-fold increase in nuclear accumulation (Fig. 4G,H and Appendix Fig. S1), indicating activation of the pathway. There was a concomitant increase in cleaved Caspase-3 (CC3) staining in those stromal cells (Fig. 4I,J). In sum, our results indicate that ablation of centrosomes in stromal progenitors causes activation of the p53-dependent mitotic surveillance pathway leading to cell apoptosis.

## Loss of Cep120 and centrosomes in the stroma causes delays in nephrogenesis

One key function of stromal progenitor cells during kidney development is the indirect regulation of nephrogenesis. Reciprocal signaling between the stromal and nephron precursors is important for the growth and specification of nephron segments (Hatini et al, 1996; Li et al, 2014). It has been shown that FoxD1+ progenitors regulate synthesis and secretion of signaling factors required for nephron progenitor proliferation and specification, as well as mesenchymal–epithelial transition (MET) (Cullen-McEwen et al, 2005; Das et al, 2013; Fetting et al, 2014; Hatini et al, 1996; Li et al, 2014; Park et al, 2007; Rowan et al, 2018; Yang et al, 2002). To

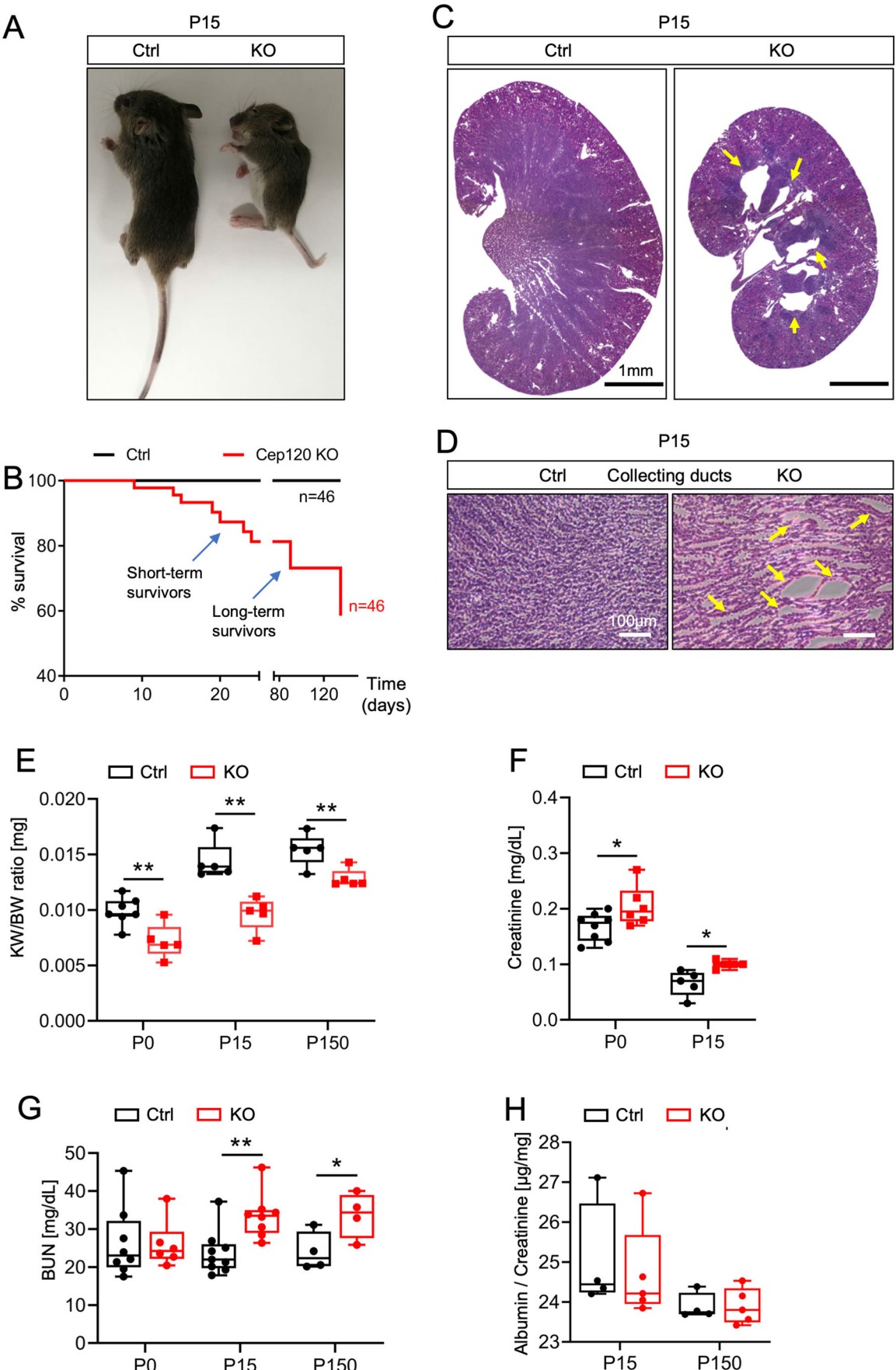

◀ **Figure 2. Loss of Cep120 and centrosomes in the stromal compartment results in small kidneys with medullary atrophy and tubular dilations.**

(A) Gross phenotype of control and Cep120-KO mice at P15. (B) Survival curve for Cep120-KO ($n = 46$) and control mice ($n = 46$). (C) Images of H&E-stained P15 kidney sections. Arrows indicate medullary atrophy in Cep120-KO kidneys. (D) Magnified images of H&E-stained P15 kidney sections with arrows highlighting collecting duct dilatations in the Cep120-KO. (E) Evaluation of kidney weight to body weight (KW/BW) ratio at different postnatal stages. (F, G) Analysis of serum creatinine and Blood Urea Nitrogen (BUN) concentration in control and Cep120-KO mice at different postnatal stages. (H) Urinary albumin to creatinine ratio in control and Cep120-KO mice at different postnatal stages. Data information: $N \geq 5$ mice per group. A two-tailed unpaired $t$ test was used for analyses and p-value denoted as follows: $*p < 0.05$, $**p < 0.01$. The vertical segments in the box plots show the first quartile, median, and third quartile. The whiskers on both ends represents the maximum and minimum for each dataset analyzed. Source data are available online for this figure.

examine whether defective centrosome biogenesis in the stromal compartment indirectly impacts nephron development, we immunostained postnatal P0 and P15 kidneys with nephron segment markers. Glomeruli were marked with antibodies against Wilms tumor 1 protein (WT1; (Mundlos et al, 1993), and the density quantified in control and Cep120-KO kidneys. The number of glomeruli did not differ significantly in the KO, suggesting that overall nephron endowment is normal (Fig. 5A,B). However, the glomeruli were more enriched in the nephrogenic zone in Cep120-KO mice compared to controls (Fig. 5A), indicating potential defects in nephron differentiation and/or maturation, or delays in overall kidney development (Hatini et al, 1996; Yang et al, 2002). Indeed, staining of proximal tubules with *Lotus tetragonolobus* lectin (LTL) and distal tubules with chloride channel-K (CLC-K) showed a decrease in abundance in KO mice (Fig. 5C,D). This is consistent with data showing that loss of stromal progenitors does not disrupt production of glomeruli per se, but does lead to defective specification and expansion of tubular epithelia in vitro (Yang et al, 2002). Moreover, in developing mouse kidneys loss of stromal progenitors severely reduces the rate of mesenchyme differentiation into a polarized tubular epithelium (Hatini et al, 1996). To test whether CL causes a delay in nephrogenesis, we analyzed kidneys collected from the long-term survivors at P150 and examined both glomeruli localization and tubule segment abundance. There were no observed changes in glomerular positioning, LTL- or CLC-K-positive tubule number (Figure EV3A–C), suggesting that centrosome loss in the stroma causes delays in nephrogenesis during embryonic kidney development that may be resolved over time.

Next, we analyzed pathways that are involved in reciprocal signaling between the stromal and nephron progenitors. It has been shown that several pathways, including Wnt/β-catenin and Hedgehog (Hh), are involved in non-cell autonomous paracrine signaling from the stroma to drive nephron maturation (Wilson and Little, 2021). Using quantitative Real Time PCR (qPCR) we examined the expression of components of Wnt/β-catenin and Hh, since the activity of these pathways are well known to rely on the centrosome-cilium complex (Anvarian et al, 2019; Goetz et al, 2009; Lancaster et al, 2011; Wong and Reiter, 2008). There was a decrease in mRNA levels of Smoothened (*Smo*, Fig. 5G) in Cep120-KO kidneys at P0, suggesting a defect in Hedgehog signaling. *Smo* deficiency in the SM results in imbalanced nephrogenic precursor formation and specification, leading to defects in tubular maturation (epithelialization) and decreased nephron numbers (Rowan et al, 2018). In addition, components of the Wnt/β-catenin signaling pathways such as *Wnt4*, *Wnt11*, *Lef1* and *Axin2* were reduced in the Cep120-KO mice (Fig. 5G), highlighting a deficiency in Wnt signaling in cells with disrupted centrosomes. This is consistent with prior studies showing that mesenchymal to epithelial transition, renal vesicle induction and proper nephron formation rely on intact reciprocal Wnt/β-catenin signaling between the stromal progenitors, metanephric mesenchyme and ureteric bud epithelia (Boivin and Bridgewater, 2018; Park et al, 2007).

In addition to regulating nephron progenitor growth and epithelialization during embryogenesis, stromal cells also play an important role in ureteric bud (UB) branching and collecting duct formation. Loss of stromal populations can result in abnormal development of the renal medulla (Hatini et al, 1996), and cause tubular dilations or cysts in adulthood (Nie and Arend, 2017). Of note, disrupting Wnt-dependent β-catenin signaling in stromal cells causes cystogenesis in collecting ducts (Boivin et al, 2015; Yu et al, 2009). We observed medullary zone atrophy and dilations of the collecting ducts within the medulla and papillary regions of Cep120-KO kidneys at P15 (Fig. 2C,D). This phenotype persisted, and became exacerbated, in the long-term survivors at 5 months of age (Fig. 5E,F). Analysis of Cep120-KO kidneys by qPCR showed decreased expression of *Wnt7b* (Fig. 5G), consistent with its role in proper cortico-medullary axis organization and development (Yu et al, 2009). In addition, both Hh and Wnt/β-catenin signaling pathways are essential for SM differentiation and pattering (Boivin and Bridgewater, 2018; Drake et al, 2020; England et al, 2020; Finer et al, 2022; Rowan et al, 2018). Reduced expression levels of Smoothened, Lef1 and Axin2, markers that are present in the SM compartment during kidney development, may be involved in the loss of stromal population upon CL. In sum, our data indicate that centrosome loss-induced changes in SM signaling result in delayed nephron maturation, aberrant medullary region specification and atrophy.

## Defective centrosome biogenesis in the stroma accelerates injury-induced fibrosis

Next, we sought to determine whether disrupting centrosome biogenesis in the renal interstitium causes fibrosis. In pathological conditions, interstitial pericytes and fibroblasts become a source of myofibroblasts that are the main drivers of fibrogenesis and extracellular matrix (ECM) deposition (Humphreys et al, 2010; Kramann et al, 2013). Previous studies have shown that defective signaling within interstitial cells can lead to spontaneous fibrosis, due to enhanced myofibroblast differentiation and activation followed by ECM deposition (DiRocco et al, 2013; Gu et al, 2017). Immunostaining of Cep120-KO kidneys at P15 with the fibrosis marker α-SMA, which marks activated myofibroblasts, showed a reduction in their levels (Fig. 3A,C). This indicates that centrosome loss alone in the SM during early development does not

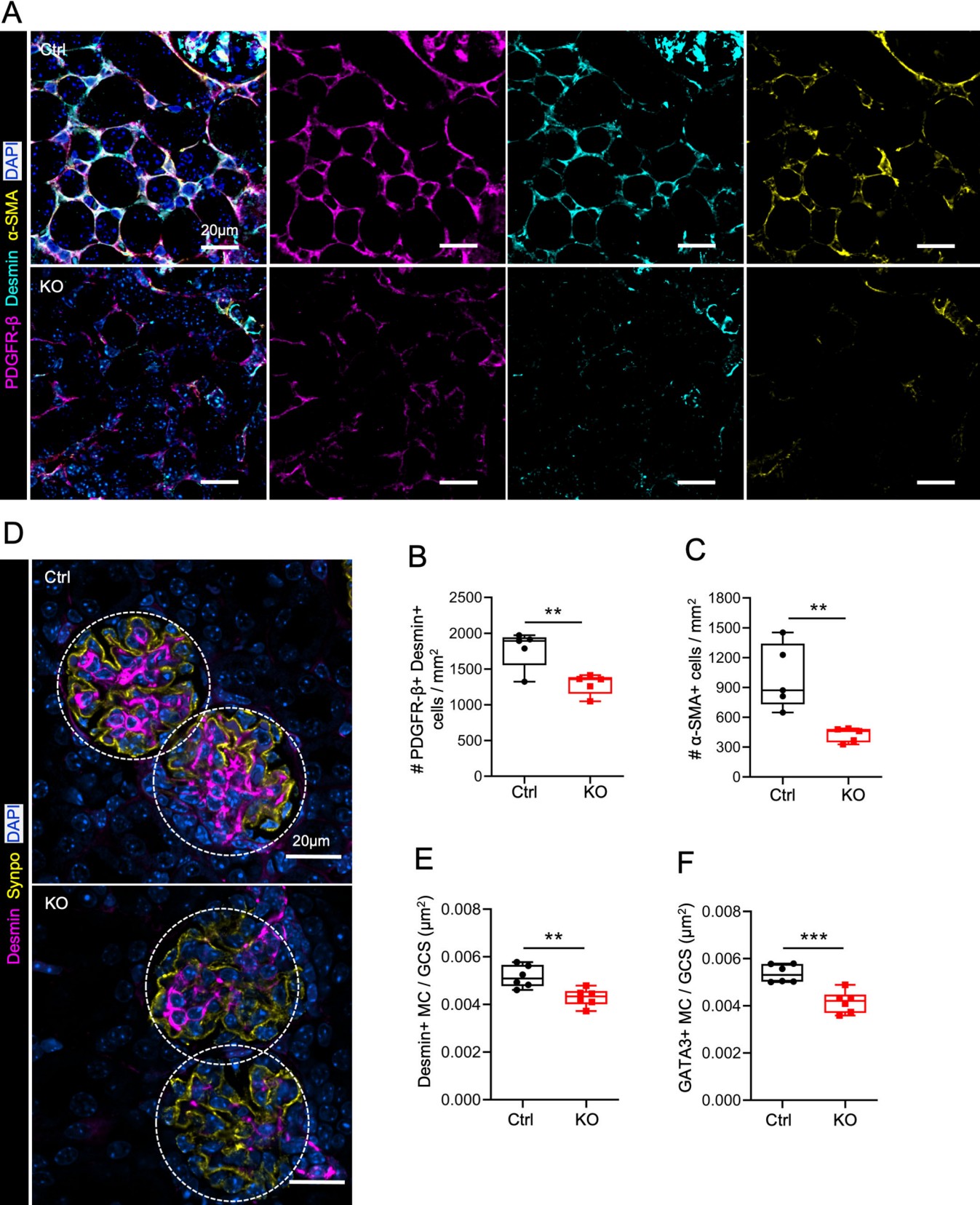

**Figure 3.  Centrosome loss results in decreased FoxD1-derived pericytes, fibroblasts and mesangial cells.**

(A) Immunofluorescence staining of control and Cep120-KO kidney sections at P15 with antibodies against PDGFR-β, desmin and α-SMA (interstitial pericytes and fibroblasts). (B, C) Quantification of pericytes and fibroblasts density expressed as number of cells per unit area. (B) $N = 1906$ cells (Ctrl) and $N = 1413$ (Cep120-KO). (C) $N = 1041$ cells (Ctrl) and $N = 461$ (Cep120-KO). (D) Immunofluorescence staining of glomeruli from control and Cep120-KO mice at P15 with antibodies against desmin (mesangial cells) and synaptopodin (podocytes). (E, F) Quantification of desmin-positive and GATA3-positive mesangial cell density per glomerular cross-sectional area. (E) $N = 1204$ cells (Ctrl) and $N = 1001$ (Cep120-KO). (F) $N = 1079$ cells (Ctrl) and $N = 817$ (Cep120-KO). Data information: $N \geq 5$ mice per group. A two-tailed unpaired $t$ test was used for analyses and $p$ value denoted as follows: $**p < 0.01$, $***p < 0.001$. The vertical segments in the box plots show the first quartile, median, and third quartile. The whiskers on both ends represents the maximum and minimum for each dataset analyzed. Source data are available online for this figure.

cause rapid fibrosis. To determine whether fibrosis develops slowly over time, we analyzed kidneys from the long-term survivors. Kidneys isolated from control and Cep120-KO mice at P150 were analyzed with markers of activated fibroblasts and ECM deposition. There was a significant increase in *Col1a1* and *Acta2* expression in Cep120-KO kidneys at P150 (Figure EV3D). Notably, there was increased expression of *Meis1*, another factor that is upregulated in kidney myofibroblasts upon injury or aging (Figure EV3D; Chang-Panesso et al, 2018). These data suggest that profibrotic and injury signatures are evident in the kidneys of long-term Cep120-KO survivors.

Next, we tested whether disrupting centrosome biogenesis in the interstitium sensitizes the kidneys and causes an enhanced fibrotic response following renal injury. We performed unilateral ureteral obstruction (UUO) in control and Cep120-KO mice at P60 and collected tissues 7 days post-injury (Fig. 6A). Analysis of the kidney injury molecule-1 (Kim1; (Humphreys et al, 2013; Ichimura et al, 1998) by qPCR and immunofluorescence showed no difference compared to injured kidneys from wildtype mice (Figure EV3E–G), indicating that the proximal tubule injury response is unaffected. However, there was elevated expression of markers of pericytes and fibroblasts (desmin, PDGFR-β, Meis1) at the RNA and protein level in the Cep120-KO kidneys upon injury (Figs. 6B,C,E and EV4A,C,D,G). Additionally, there was upregulation of markers of fibrosis and ECM deposition (fibronectin, α-SMA, collagen 1; Figs. 6B,D,F and EV4B,E,F).

Finally, we sought to determine the mechanisms by which centrosome loss may be causing the observed fibrosis phenotypes post-injury. The glioma-associated oncogene (*Gli*) transcription factors, effectors of Hedgehog signaling, are known to regulate myofibroblast proliferation and fibrosis (Fabian et al, 2012; Kramann et al, 2015). Specifically, *Gli2* drives cell cycle progression of myofibroblasts during renal fibrosis (Kramann et al, 2015). Analysis of kidneys isolated at day 7 post-injury showed elevated expression of *Gli2*, but not *Gli1*, at the mRNA and protein levels (Figs. 6B,G and EV5A–C). Gli2 activity can be regulated in a Hh ligand-dependent mechanism, and in a ligand-independent way via TGF-β/Smad signaling (Fig. 6H; Dennler et al, 2007; Meng et al, 2015). qRT-PCR analysis of UUO kidneys showed no change in the expression levels of the two main Hh ligands, Sonic hedgehog (*Shh*) and Indian hedgehog (*Ihh*) (Figs. 6I and EV5D). Similarly, there was no difference in the expression of Patched1 (*Ptch1*) receptor (Figure EV5E), suggesting that activation of Gli2 may be Hh ligand-independent. In contrast, there was a significant increase in *Tgfb1* expression in the injured Cep120-KO kidneys (Fig. 6J), indicating that this pro-fibrotic pathway is likely responsible for the increased Gli2 activation upon centrosome loss. In support of this theory, there was elevated expression of *Smad3* transcripts in Cep120-KO

kidneys (Fig. 6K), and a concurrent increase in phosphorylated Smad3 (Figs. 6B and EV5F). Overall, these data indicate that disrupting centrosomes in the renal interstitium accelerates injury-induced fibrosis via the ligand-independent TGF-β/Smad3-Gli2 axis.

## Discussion

In this study, we examined the consequences of defective centrosome biogenesis in stromal progenitors during embryonic kidney development. We discovered that ablating Cep120 caused centrosome loss in FoxD1+ progenitors and their derived lineages. This resulted in abnormal kidney development and size, medullary atrophy, and tubular dilations over time. However, analysis of BUN and serum creatinine levels indicated that kidney filtration function was unaffected, and stayed within physiological range. Therefore, the early lethality observed in Cep120-KO mice is likely due to extrarenal phenotypes, as FoxD1 is also expressed in anterior hypothalamus, retinal ganglion, ventral diencephalon and lung pericytes (Carreres et al, 2011; Herrera et al, 2004; Hung et al, 2013; Newman et al, 2018). This lethality has been previously reported when ablating genes using FoxD1-Cre-expressing mice (Karolak et al, 2018; Nie and Arend, 2017). The cell-autonomous effect of CL in stromal progenitors resulted in reduced abundance of SM cells, interstitial pericytes and fibroblasts, as well as mesangial cells. In a non-cell autonomous fashion, CL in the stromal compartment caused a delay in nephron development and maturation, which was resolved over time in the long-term Cep120-KO survivors. However, the tubular dilations and medullary atrophy persisted. Centrosome loss in stromal progenitors did not result in spontaneous fibrosis and ECM deposition, although some profibrotic and injury signatures were evident in the long-term survivors at 5 months of age. Importantly, inducing injury in Cep120-KO mice showed that CL in the stroma plays a key role in accelerating injury-induced fibrosis.

How does Cep120 and centrosome loss in stromal progenitors result in small, dysplastic kidneys? We found that the population of SM cells and their derived lineages were significantly reduced in Cep120-KO kidneys (Figs. 3 and 4A,C), and propose that this is caused by a combination of CL-induced cell cycle delay and apoptosis, as well as abnormal expression of signaling pathways involved in stromal cell growth and differentiation. Changes in these pathways, including Hh and Wnt (Fig. 5G), can lead to both cell-autonomous and non-autonomous defects in reciprocal signaling between the stromal mesenchyme and nephron tubule compartments during embryonic kidney development (Wilson and Little, 2021). We observed that CL in stromal progenitors results in

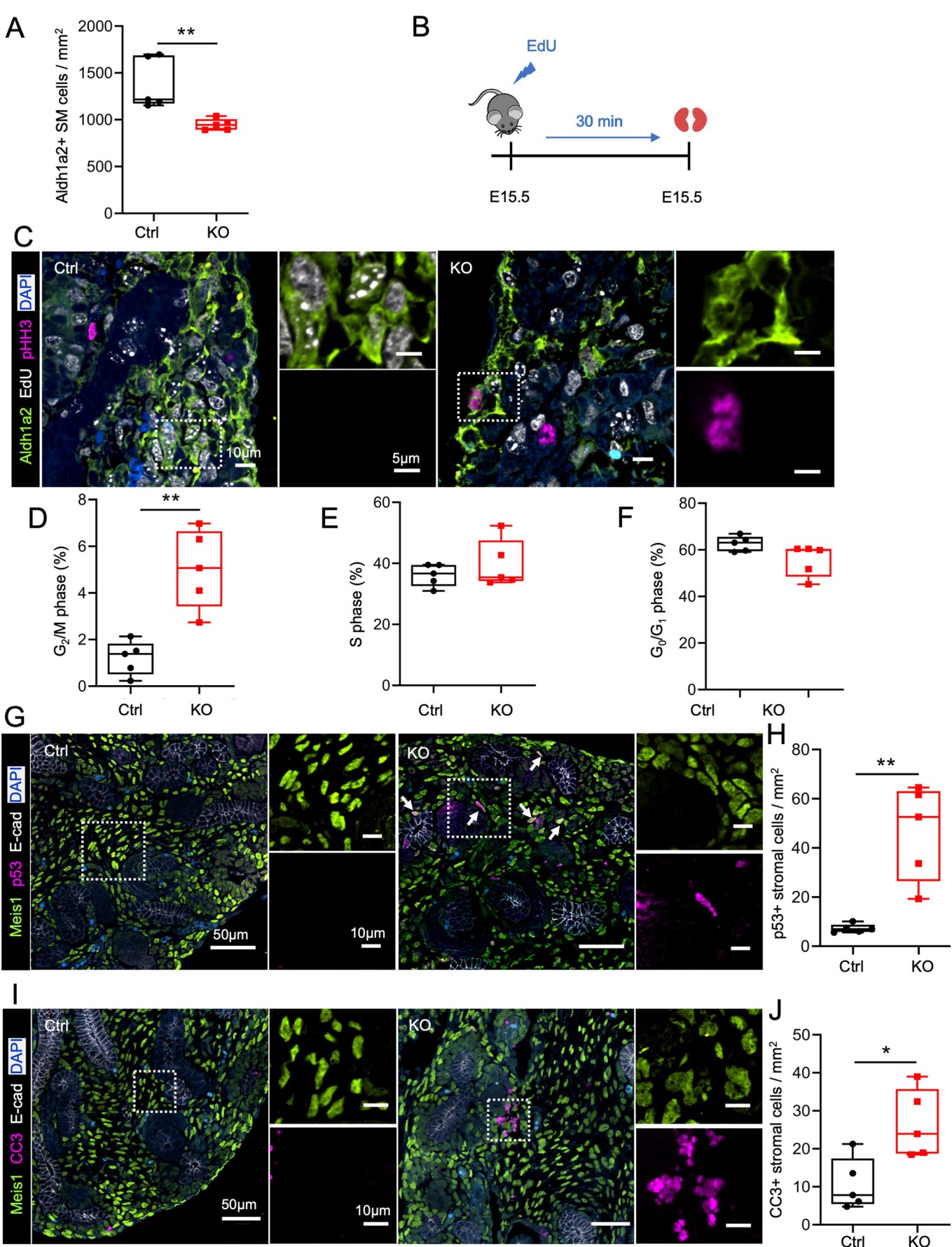

◄ **Figure 4.  Centrosome loss causes a $G_2$/M delay and promotes apoptosis of stromal progenitors.**

(A) Quantification of Aldh1a2-positive SM cells at E15.5. The density is expressed as number of cells per unit area. $N = 1450$ cells (Ctrl) and $N = 1199$ (Cep120-KO). (B) Schematic of EdU (5-ethynyl-2′-deoxyuridine) injection protocol. Control and Cep120-KO pregnant females were injected at day 15.5 post-fertilization, followed by kidney isolation after 30 minutes. (C) Immunofluorescence staining of E15.5 kidney sections for SM cells (Aldh1a2), EdU (replicating DNA) and pHH3 (cells in $G_2$ and mitosis). (D–F) Quantification of SM cells at each phase of the cell cycle; (D) $G_2$M phase (Aldh1a2+/pHH3+/EdU−); (E) S-phase (Aldh1a2+/EdU+/pHH3−); (F) $G_0/G_1$ (Aldh1a2+/EdU−/pHH3−). $N = 1450$ cells (Ctrl) and $N = 1199$ (Cep120-KO). (G) Immunofluorescence staining of E15.5 kidney sections with antibodies against interstitial stroma cells (Meis1) and p53. (H) Quantification of the percentage of Meis1-positive cells expressing p53. $N = 3280$ cells (Ctrl) and $N = 2746$ (Cep120-KO). (I) Immunofluorescence staining of E15.5 kidney sections with antibodies to mark interstitial stroma cells (Meis1) and cleaved Caspase 3 (CC3). (J) Quantification of Meis1-positive cells expressing cleaved Caspase 3 per unit area. $N = 3272$ cells (Ctrl) and $N = 2698$ (Cep120-KO). Data information: $N = 5$ mice per group. A two-tailed unpaired $t$ test was used for analyses and $p$ value denoted as follows: $*p < 0.05$, $**p < 0.01$. The vertical segments in the box plots show the first quartile, median, and third quartile. The whiskers on both ends represents the maximum and minimum for each dataset analyzed. Source data are available online for this figure.

$G_2$M phase cell cycle delay, which was followed by nuclear accumulation of p53 protein and caspase-mediated apoptosis (Fig. 4G–J). This is consistent with previous observations in several organs and species showing activation of the mitotic surveillance pathway in cells upon defective centrosome biogenesis (Fong et al, 2016; Lambrus et al, 2016; Mikule et al, 2007; Phan et al, 2021; Wong et al, 2015; Xie et al, 2021). In the developing mouse brain, disruption of centrosome biogenesis leads to prolonged mitosis, p53-mediated apoptosis, depletion of the neural progenitor pool, and leads to the microcephaly phenotype (Phan et al, 2021). Similarly, CL in the developing mouse lung causes apoptosis of Sox2-expressing airway progenitor cells (Xie et al, 2021). Thus, one mechanism leading to reduced abundance of SM progenitors upon CL is likely due to cell death.

Another mechanism that can lead to depletion of SM-derived cell populations is the abnormal differentiation/fate determination of the progenitor cells. For example, CL in neural progenitor cells of *Drosophila* and mice disrupts asymmetric cell division, results in imbalanced distribution of fate determinants, and causes premature differentiation into neurons. This ultimately depletes the neural stem cell pool and leads to the small brain phenotype (Homem et al, 2015; Robinson et al, 2020; Wang et al, 2009). Our experiments identified both Hedgehog and Wnt/β-catenin pathway components to be downregulated upon centrosome loss (Fig. 5G). Both pathways play important cell-autonomous roles in regulating SM cell proliferation and differentiation. Ablation of Hedgehog effector Smoothened in FoxD1+ stromal progenitors results in reduced proliferation and increased apoptosis, leading to almost complete absence of the capsular stromal layer (Rowan et al, 2018). β-catenin deficiency in FoxD1+ progenitors induces apoptosis and blocks their differentiation into medullary stromal cells (Boivin and Bridgewater, 2018; England et al, 2020). Moreover, both signaling pathways exert non-cell autonomous effects on nephron tubular epithelial growth, differentiation and cortico-medullary axis formation (Drake et al, 2020; England et al, 2020; Rowan et al, 2018; Yu et al, 2009). Ablation of Smoothened in FoxD1+ stromal progenitors reduces formation of nephrogenic precursor structures with an accompanying decrease in overall nephron number (Rowan et al, 2018). Activation of β-catenin in the stromal lineage non-autonomously prevents the differentiation of nephron progenitor cells (Drake et al, 2020). Since the centrosome-cilium complex is known to regulate both of these pathways, CL in the SM may disrupt the reciprocal signaling between the nephrogenic and stromal compartments, leading to the observed delays in nephron segment maturation and tubular dilations (Fig. 5A–F). Altogether, we conclude that the reduction in stromal progenitors and derived

lineages upon CL results from a combination of cell cycle delay, apoptotic cell death, and defects in cell-autonomous and paracrine signaling.

In contrast to the stromal progenitor cells, mature interstitial cell lineages (such as pericytes and myofibroblasts) were still able to proliferate in the absence of centrosomes following kidney injury (Figs. 6 and EV4). One possibility is that components of the p53-mediated mitotic surveillance pathway may be differentially expressed in SM compared to the differentiated stromal cells, leading to preferential activation only in the progenitors. Another potential mechanism is the expression of molecules that confer a protective role against CL-induced cell death. For example, high levels of pERK1/2 in lung or intestinal progenitor cells provides protection against CL-induced cell cycle arrest and apoptosis (Xie et al, 2021). High ERK activity is essential for progression of renal fibrosis by promoting detrimental differentiation and expansion of kidney fibroblasts, including in models of unilateral ureteral obstruction (Andrikopoulos et al, 2019; Rodríguez-Peña et al, 2008). Intriguingly, previous single-cell RNAseq studies showed that expression levels of ERK1/2 increase in fibroblasts and myofibroblasts following UUO (Li et al, 2022). Thus, it is possible the proliferative potential of centrosome-less pericytes and myofibroblasts upon UUO is driven by high levels of ERK1/2 activity, which serves as a protective factor for CL-induced cell cycle arrest and apoptosis.

Our data demonstrate that the injury-induced fibrotic response, including pericyte proliferation, myofibroblast activation and ECM deposition, is enhanced upon CL. We hypothesize this is due to abnormal TGF-β/Smad3-Gli2 signaling. Gli2 drives cell cycle progression of myofibroblasts during renal fibrosis (Kramann et al, 2015), and its activity can be regulated either in a Hh ligand-dependent mechanism, or ligand-independent way via TGF-β/Smad signaling (Fig. 6H; Dennler et al, 2007; Meng et al, 2015). Our analysis of UUO kidneys showed no change in the expression levels of the two main Hh ligands (*Shh* and *Ihh*) or the Patched1 receptor. In contrast, there was a significant increase in *Tgfb1* expression and Smad3 activation in the injured Cep120-KO kidneys (Figs. 6 and EV5), indicating that this pro-fibrotic pathway is likely responsible for the increased Gli2 activation upon centrosome loss. TGF-β/Smad signaling depends on an intact cilium-centrosome complex, with TGF-β receptors localized to the ciliary tip and endocytic vesicles at the ciliary base (Clement et al, 2013; Vestergaard et al, 2016). Upon ligand stimulation, TGF-β receptors localize at the ciliary pocket and activate Smad3, which then translocates to the nucleus to activate transcription factors

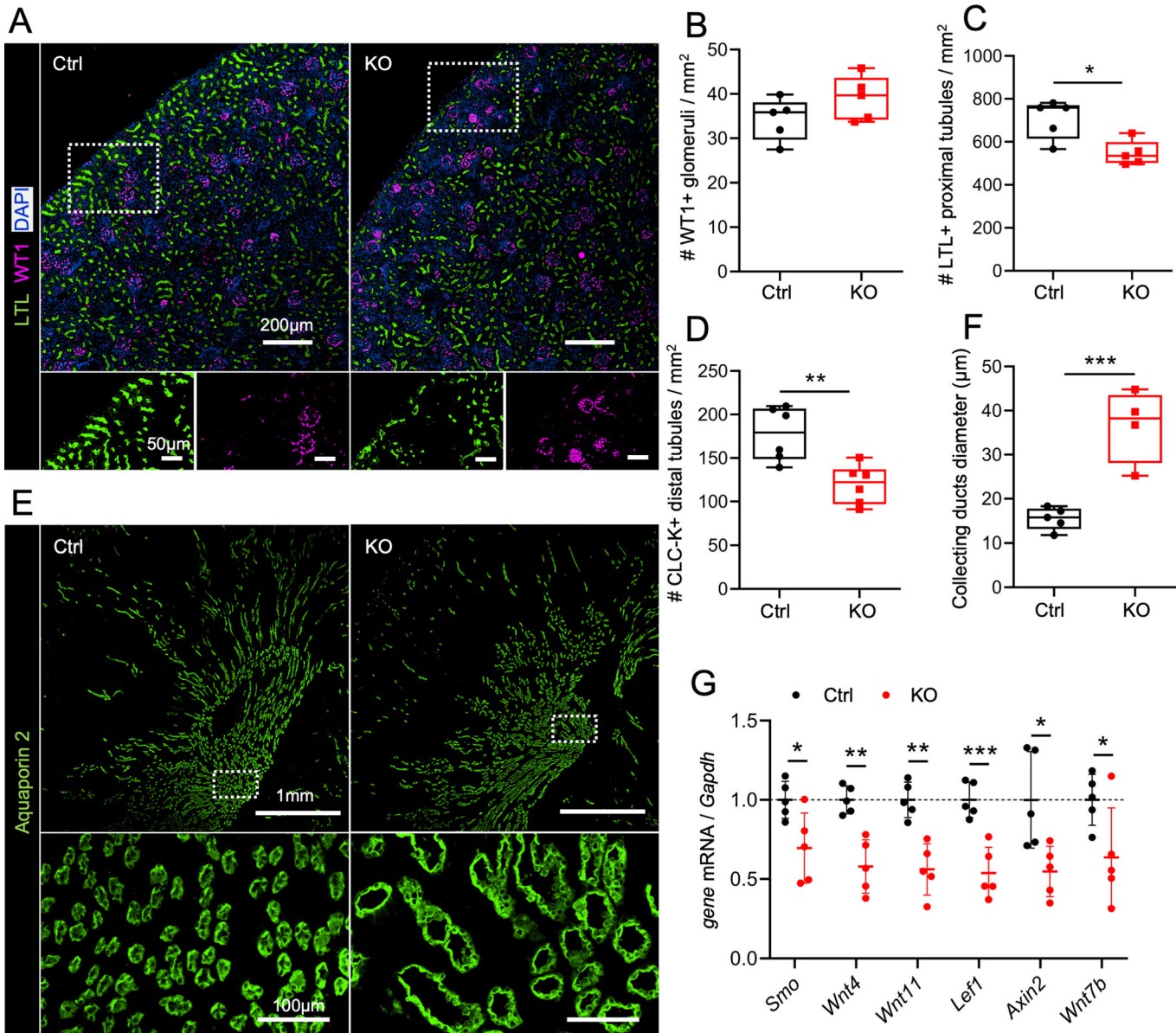

**Figure 5. Cep120 and centrosome loss in stromal cells causes delayed nephron maturation.**

(A) Immunofluorescence staining of P15 control and Cep120-KO kidney sections with antibodies to mark glomeruli (WT1) and proximal tubules (LTL). (B–D) Quantification of (B) WT1-positive glomeruli, (C) LTL-positive proximal tubules and (D) CLC-K-positive distal tubules per unit area. (B) $N = 1306$ glomeruli (Ctrl) and $N = 1128$ (Cep120-KO). (C) N = 3889 tubules (Ctrl) and $N = 3859$ (Cep120-KO). (D) $N = 727$ tubules (Ctrl) and $N = 507$ (Cep120-KO). (E) Immunofluorescence staining of P150 kidney sections with antibodies to mark collecting duct tubules (Aquaporin 2). (F) Quantification of Aquaporin 2-positive collecting duct tubule diameter. $N = 256$ tubules (Ctrl) and $N = 140$ (Cep120-KO). (G) qPCR-based quantification of the relative change in gene expression levels of Smo, Wnt4, Wnt11, Lef1, Axin2 and Wnt7b from P0 kidneys (expressed as fold change). Data information: $N = 5$ mice per group. A two-tailed unpaired $t$ test was used for analyses and $p$ value denoted as follows: $^*p < 0.05$, $^{**}p < 0.01$, $^{***}p < 0.001$. The vertical segments in the box plots (B–D) show the first quartile, median, and third quartile. The whiskers on both ends represents the maximum and minimum for each dataset analyzed. Data in (G) are represented as mean ± SEM. Source data are available online for this figure.

(Clement et al, 2013; Vestergaard et al, 2016). Cilia are also present in kidney pericytes and fibroblasts in culture, however spontaneous loss of cilia is observed concurrent with their TGF-β-induced transition into myofibroblasts, which can be inhibited upon ablation of ciliary genes (Jung et al, 2022; Rozycki et al, 2014). Intriguingly, it has been shown that phosphorylated Smads can localize to centrosomes and undergo ubiquitin-dependent

proteasomal degradation (Fuentealba et al, 2008; Fuentealba et al, 2007). Therefore, one way CL could increase levels of TGF-β/Smad pathway components is via inhibition (or elimination) of sites of their degradation. Overall, we interpret our data to suggest that disrupting centrosomes in the renal interstitium accelerates injury-induced fibrosis via defective ligand-independent TGF-β/Smad3-Gli2 signaling axis.

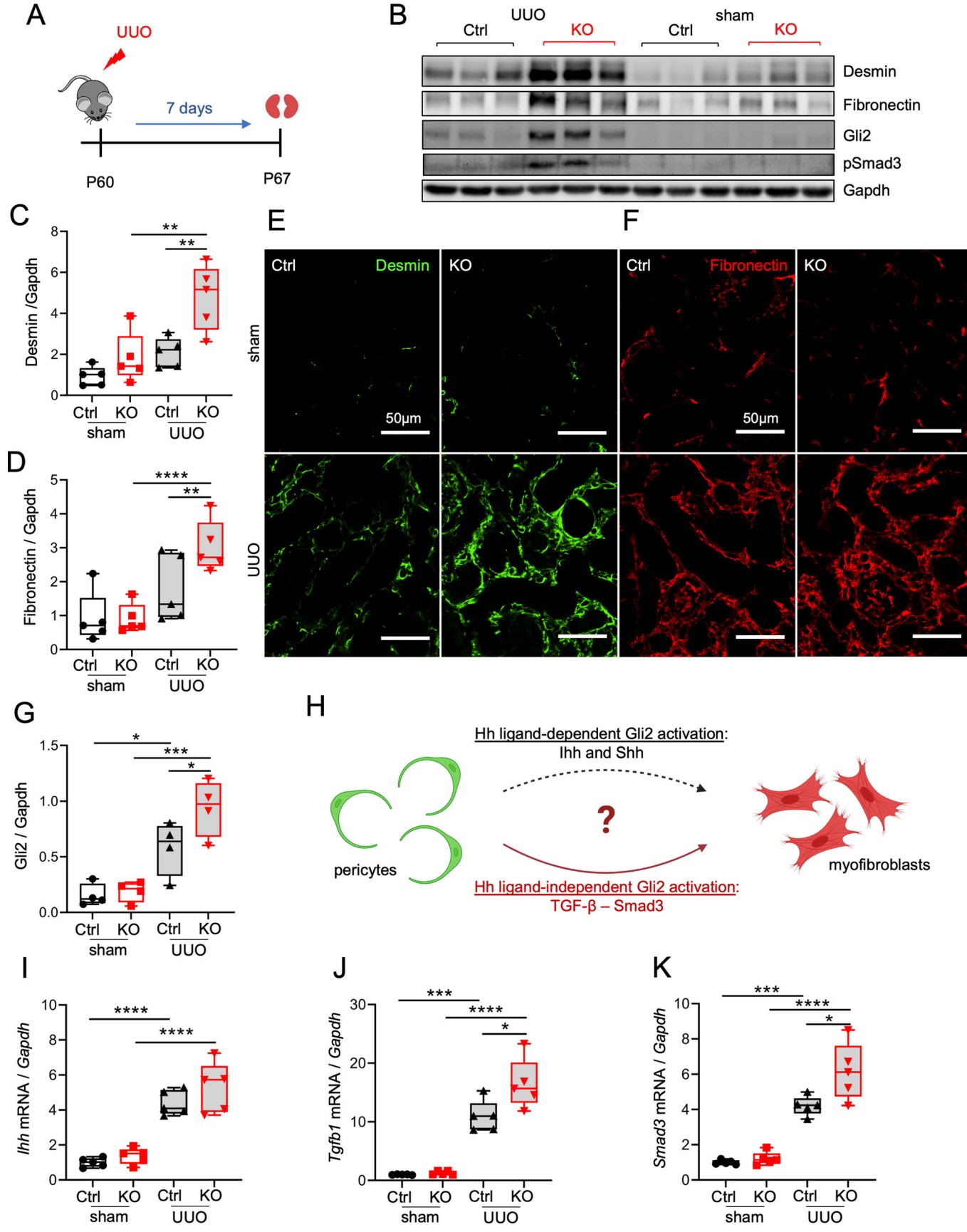

**Figure 6. Defective centrosome biogenesis in the stroma accelerates injury-induced fibrosis.**

(A) Schematic of unilateral ureteral obstruction (UUO) injury model. UUO was performed on control and Cep120-KO mice at P60 and kidneys isolated after 7 days. (B) Immunoblot analysis of lysates from control and Cep120-KO kidneys after 7 days of injury (UUO; left) and without injury (sham; right). (C, D) Densitometry analysis of (C) desmin and (D) fibronectin normalized to Gapdh. Data are expressed as fold change relative to the sham control. (E, F) Immunofluorescence images of UUO kidneys from control and Cep120-KO mice stained with antibodies to mark pericytes (desmin) and extracellular matrix deposition (fibronectin). (G) Densitometry analysis of Gli2 normalized to Gapdh. Data are expressed as fold change relative to the sham control. (H) Schematic of pericytes-to-myofibroblasts transition highlighting the two potential pathway mechanisms. Differentiation to myofibroblasts may occur via Hh ligand-dependent or independent pathways. (I–K) qPCR-based quantification of the relative change in gene expression levels of (I) Indian hedgehog (Ihh), (J) Tgfb1, and (K) Smad3 in control and Cep120-KO kidneys after 7 days of injury (UUO) and without injury (sham), expressed as fold change relative to the sham-control. Data information: $N = 5$ mice per group. A one-way ANOVA test followed by multiple-group comparison analysis with Tukey correction was used for analyses, and $p$ value denoted as follows: $*p < 0.05$, $**p < 0.01$, $***p < 0.001$, $****p < 0.0001$. The vertical segments in the box plots show the first quartile, median, and third quartile. The whiskers on both ends represents the maximum and minimum for each dataset analyzed. Source data are available online for this figure.

How do mutations in Cep120 and other centrosome biogenesis proteins that manifest in the stromal progenitors contribute to the overall renal dysplasia, fibrotic scarring and cyst formation that occurs in NPH, JS, and JATD patients? In a concurrent study (Cheng et al, 2023) our group also examined the consequences of Cep120 loss in the nephron progenitor and collecting duct niches of the developing kidney. Conditional deletion of Cep120 in the nephron progenitor cells and collecting duct progenitors similarly led to abnormal development and small dysplastic kidneys at birth. Defective centrosome biogenesis in these progenitors also caused delayed mitosis, increased cell death, and premature differentiation, which correlated with changes in Wnt signaling. This indicates that aberrant centrosome biogenesis in all three kidney progenitor populations is responsible for the small dysplastic kidney phenotype in patients. In contrast to CL in FoxD1-positive progenitors, ablation of Cep120 in the CM and UB progenitors resulted in rapid and progressive cystogenesis (Cheng et al, 2023) and significant kidney function decline. These phenotypes were not observed in the FoxD1-Cep120 model. Therefore, we propose that mutations in Cep120 and centrosome biogenesis genes in the nephron progenitors play a major role in cystogenesis, while defects in stromal progenitors contribute to the congenital developmental defects in the patients. In addition, defective centrosome biogenesis in stromal progenitors sensitizes the kidneys to injury and contributes to the enhanced fibrosis seen in the fibrocystic kidneys of patients. In sum, our study provides a detailed characterization of the underlying molecular and cellular defects in renal centrosomopathies.

## Methods

### Animal experiments

All animal studies were performed according to the guidelines of the Institutional Animal Care and Use Committee at Washington University (protocol approval # 21-0252) and the National Institutes of Health. Generation of the Cep120$^{F/F}$ mice has been described previously (Wu et al, 2014). Conditional deletion of Cep120 in stromal mesenchyme progenitor cells was induced by crossing Cep120$^{F/F}$ mice with a FoxD1-(GFP)-Cre (FoxD1$^{GC/+}$; JAX stock no. 012463) (Humphreys et al, 2010) transgenic strain. For timed pregnancies, the day of the copulatory plug was defined as E0.5. Kidneys were isolated from experimental (Cep120$^{F/F}$; FoxD1$^{GC/+}$) and control (Cep120$^{F/+}$; FoxD1$^{GC/+}$ or Cep120$^{F/F}$; FoxD1$^{+/+}$) animals at E15.5 of embryonic development and postnatally (P0, P15, and P150). List of primers used for genotyping can be found in Table EV2.

Genotyping was performed following established protocols from the donating investigators (Wu et al, 2014) and the Jackson Laboratory.

For quantification of kidney size, the animal body weight was measured just before euthanasia, and the kidney weight shortly after harvesting. The kidney weight to body weight ratio was calculated by dividing the total weight of the left and right kidney by the body weight. For analysis of renal function, Blood Urea Nitrogen (BUN) levels were assessed using the QuantiChrom Urea Assay Kit (DIUR-100, BioAssay Systems) according to manufacturer's protocol. Serum creatinine levels were measured by HPLC at the O'Brien Core Center for Acute Kidney Injury Research (University of Alabama School of Medicine, Birmingham, Alabama, USA). For analysis of proteinuria, urine creatinine levels were quantified using QuantiChrom Creatinine Assay Kit (DIUR-500, BioAssay Systems). Creatinine concentrations were used to normalize the amounts of urine analyzed by SDS-PAGE and Coomasie Brilliant Blue staining. Gels after de-staining were imaged using a ChemiDoc™ MP Imaging System (Bio-Rad). Albumin concentration was determined by densitometry and comparison to BSA standards loaded on the same gel. Statistical significance between groups was assessed and data were presented as albumin to creatinine ratio.

### Tissue preparation and histology

Isolated kidneys were fixed overnight in 4% paraformaldehyde in PBS at 4 °C and then submitted for paraffin embedding. Tissues were cut into 7.5-µm thick sections using a microtome (RM2125 RTS; Leica) and collected onto Superfrost slides (Thermo Fisher Scientific). For histological assessment, sections were stained with hematoxylin and eosin (H&E) according to standard protocols.

### Immunohistochemistry

For antigen unmasking, tissues were incubated in TE antigen-retrieval buffer (10 mM Tris Base, 1 mM EDTA, and 0.05% Tween-20, pH 9.0) and boiled for 30 minutes. Samples were permeabilized with 0.3% Triton X-100 in PBS for 10 min, and blocked using 3% BSA in PBS-T (PBS with 0.1% Triton X-100) for 1 h. Incubation with primary antibodies was performed overnight at 4 °C, followed by three washes with PBS-T, a 1-h incubation with Alexa Fluor dye-conjugated secondary antibodies (1:500) at room temperature, then counterstained with DAPI for 5 min. The full list of antibodies used is provided in Table EV3. Images from fixed tissue sections were captured using a Nikon Eclipse Ti-E inverted confocal microscope equipped with a ×40 (1.3NA) and ×60 (1.4NA) Plan Fluor oil

immersion objective lens (Nikon, Melville, NY). A series of digital optical sections (Z-stacks) were captured between 0.3 and 1.5 μm intervals using a Hamamatsu ORCA-Fusion Digital CMOS camera.

## Evaluation of Cep120 and centrosome loss

For quantification of Cep120 and centrosome loss, kidney samples were stained with antibodies against Cep120, Centrin, Cep135, Ninein and γ-tubulin (to mark centrosomes and centrioles). Sections were co-stained with antibodies to identify specific cells of interest: Aldh1a2 (stromal mesenchymal cells), PDGFR-β (pericytes and fibroblasts), GATA3 and desmin (mesangial cells), and α-SMA (vascular smooth muscle cells). In all, 8–12 random fields were captured with a ×40 objective lens throughout the kidney to locate respective cell types. Normal centrosome number was defined as cells containing one or two foci of γ-tubulin, and the fraction of cells containing zero Cep120 and γ-tubulin foci was defined as centrosome loss.

## Cell cycle analysis

For cell cycle analysis, pregnant females were injected with EdU (50 mg/kg) and sacrificed after 30 min for embryo collection. E15.5 embryonic kidneys were sectioned and stained according to the protocol of the Click-iT™ EdU Imaging Kit (Invitrogen by Thermo Fisher Scientific). Sections were co-stained with antibodies against phospho-Histone H3 (pHH3), and 8–12 random fields were captured with a ×40 objective within cortical region. A series of digital optical sections (Z-stacks) were captured between 0.3 and 1.5 μm intervals. Quantification of Aldh1a2-positive stromal mesenchymal cells in each phase of the cell cycle was performed by counting EdU-/pHH3- cells ($G_0/G_1$), EdU+ cells (S-phase), and pHH3+ ($G_2/M$) cells in each Z-section, which were then expressed as a percentage of all Aldh1a2+ cells counted.

## Analysis of cell lineages derived from stromal progenitors

Quantification of interstitial fibroblasts was performed by staining of P15 kidney sections with three markers of these cells: PDGFR-β, desmin and α-SMA. 10–15 random fields in cortical and medullary regions of each kidney were captured with a 40x objective lens. PDGFR-β and desmin double-positive cells as well as α-SMA-positive cells were counted, and expressed as number of cells per unit area. For glomerular mesangial cell quantification, P15 kidney sections were stained with antibodies against desmin, GATA3 and α-SMA and counterstained with podocyte marker, synaptopodin. The number of desmin-, GATA3- or α-SMA-positive mesangial cells were scored in at least 12 glomeruli per each kidney section, and represented as the number of cells per glomerular cross-sectional (GCS) area. The GCS area of each glomerulus was measured by outlining the Bowman's capsule with the "area measurement" tool in ImageJ. α-SMA expression by pericytes surrounding the glomeruli was quantified by measuring the area stained by α-SMA expressed as percentage of the GCS area.

## Evaluation of nephrogenesis defects

Quantification of glomeruli number was performed in kidney sections stained with antibodies against podocyte marker WT1

(Mundlos et al, 1993). The total number of glomeruli was quantified per section and expressed per unit area. For tubular counts, kidney sections were stained with proximal tubules brush border marker, LTL (Chevalier et al, 1998), and antibodies against distal tubule marker, CLC-K (Kieferle et al, 1994). In all, 10–15 random fields were captured with a ×40 objective, the number of tubules counted and expressed per unit area. Evaluation of collecting ducts dilation was performed in kidney sections of wildtype and Cep120-KO mice isolated at P150, and stained with antibodies against aquaporin 2. At least 10 random fields were captured from each section and the diameter of each tubular cross-section was measured using the Nikon Elements AR 5.21 length measurement tool.

## Unilateral ureteral obstruction experiments

Unilateral ureteral obstruction (UUO) was performed on adult (8–12-week-old) mice as previously described (Kefaloyianni et al, 2016). Briefly, after flank incision, the left ureter was tied off at the level of the lower pole with two 3.0 silk ties, and left in place for 7 days. Sham-operated mice underwent the same surgical procedure, except for the ureter ligation step. Seven days following surgery, mice were sacrificed, and the obstructed and contralateral non-obstructed kidneys were harvested for analysis.

For quantification of tubular injury, UUO and sham kidney samples were stained with antibodies against the kidney injury molecule 1 (Kim1). Multiple images from each kidney section were acquired and the number of Kim1-positive tubules was scored per unit area. For quantification of fibroblast expansion and fibrosis in UUO and sham kidney samples, the sections were stained with antibodies against desmin, PDGFR-β, α-SMA and fibronectin. At least 10 fields of each kidney section were acquired, and the relative area stained was calculated using ImageJ software.

## RNA isolation, RT-PCR, and quantitative PCR

Total RNA was isolated from mouse kidneys using Direct-zol™ RNA MiniPrep Plus (Zymo Research). 1 μg of RNA was reverse-transcribed using a High Capacity cDNA Reverse Transcription Kit (Applied Biosystems by Thermo Fisher Scientific) according to the manufacturer's protocol. Real-time PCR was performed with SYBR Select Master Mix (Applied Biosystems by Thermo Fisher Scientific) in a 96- or 384-well plate format. 50 ng of cDNA was used in 10 μl final volume and reactions run at the standard cycling mode described by manufacturer, using QuantStudio 6 Flex system (Applied Biosystems). *Gapdh* was used as endogenous control and data analyzed with ΔΔCt method. Primers used are listed in Table EV4.

## Protein isolation and Western blot

Kidneys were collected from mice at 2 months of age following UUO surgery, snap-frozen in liquid nitrogen and homogenized in ice-cold RIPA Buffer (1% NP40, 0.5% sodium deoxycholate, 0.1% SDS, 2 mM EDTA, 1 mM $Na_3VO_4$, 20 mM NaF, 0.5 mM DTT, 1 mM PMSF, and protease inhibitor cocktail in PBS, pH 7.4). The homogenized lysates were incubated for 1 h on ice, centrifuged at 12,000xg for 10 min at 4 °C and the supernatants collected. Total

protein concentration was assessed using the Pierce™ BCA Protein Assay Kit (Thermo Scientific). 20–40 μg of total protein was separated on 7–10% SDS-PAGE polyacrylamide gels and transferred onto Immobilon PVDF membranes (Millipore). Membranes were blocked using 5% non-fat dry milk in TBS supplemented with 0.1% Tween-20 for 1 h at room temperature. Blots were incubated overnight at 4 °C with primary antibodies. Horseradish peroxidase-conjugated secondary antibodies (1:5000) in TBS-T with 5% milk were added to membranes and incubated for 2 h. After several washes, the membranes were incubated with enhanced chemiluminescence detection substrate (Thermo Scientific) and proteins visualized using a ChemiDoc™ MP Imaging System (Bio-Rad). Full list of antibodies used is provided in Table EV3.

## Statistical analysis

Statistical analyses were performed using GraphPad PRISM 9.0. Data were analyzed with One-way ANOVA followed by multiple-group comparison analysis with Tukey correction, or two-tailed unpaired $t$ test. The vertical segments in box plots show the first quartile, median, and third quartile. The whiskers on both ends represent the maximum and minimum values for each dataset analyzed. A $p$ value $< 0.05$ was considered statistically significant and denoted as follows: $*p < 0.05$, $**p < 0.01$, $***p < 0.001$, $****p < 0.0001$. Experiments were not blinded.

# Data availability

This study includes no data deposited in external repositories.

# Peer review information

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

## Acknowledgements

We would like to thank Dr. Sanjay Jain, Dr. Jeff Miner, Dr. Hani Suleiman, Dr. Ben Humphreys, Dr. Andreas Herrlich, Dr. Michael Rauchman, Dr. Maggie Chen, and Dr. Feng Chen (all at Washington University in St Louis) for sharing reagents used in this study. We also thank members of the Mahjoub lab and the Washington the University Ciliopathy Research Group for helpful advice and feedback on this project, as well as critical reading of the manuscript. We acknowledge support from the UAB-UCSD O'Brien Core Center for Acute Kidney Injury Research (NIH P30-DK079337) for analysis of serum-creatinine levels for this project. Several figures were created with the use of BioRender. This study was supported by funding from the National Institute of General Medical Sciences (R01GM140115) to BW and the National Institute of Diabetes and Digestive and Kidney Diseases (R01DK108005) to MRM.

## Author contributions

**Ewa Langner**: Conceptualization; Formal analysis; Validation; Investigation; Visualization; Writing—original draft; Writing—review and editing. **Tao Cheng**: Formal analysis; Investigation. **Eirini Kefaloyianni**: Resources; Methodology. **Charles Gluck**: Formal analysis; Investigation. **Baolin Wang**: Resources. **Moe R Mahjoub**: Conceptualization; Resources; Data curation; Supervision; Funding acquisition; Project administration; Writing—review and editing.

## Disclosure and competing interests statement

The authors declare no competing interests.

# Expanded View Figures

**Figure EV1.   Cep120 depletion in stromal progenitors causes centrosome loss in the derived cell types.**                                                               ▶

(A) Immunofluorescence staining of kidney sections from control and Cep120-KO mice at P15 with antibodies to mark Cep120, centrosomes (γ-tubulin) and pericytes (PDGFR-β). (B, C) Quantification of the percentage of PDGFR-β-positive pericytes with (B) Cep120 expression and (C) centrosomes at P15. $N = 680$ cells (Ctrl) and $N = 421$ (Cep120-KO). (D) Immunofluorescence staining of P15 kidney sections with antibodies to mark Cep120, mesangial cells (GATA3) and podocytes (synaptopodin). (E, F) Quantification of the percentage of GATA3-positive mesangial cells expressing (E) Cep120 and (F) Cep135 (centrosomes) at P15. (E) $N = 1079$ cells (Ctrl) and $N = 817$ (Cep120-KO). (F) $N = 720$ cells (Ctrl) and $N = 696$ (Cep120-KO). (G, H) Quantification of the percentage of α-SMA-positive vascular smooth muscle cells (VSMC) expressing (G) Cep120 and (H) Cep135 at P15. (G) $N = 1025$ cells (Ctrl) and $N = 881$ (Cep120-KO). (H) $N = 576$ cells (Ctrl) and $N = 588$ (Cep120-KO). (I, J) Quantification of the percentage of CD31-positive endothelial cells (EC) expressing Cep120 (I) and Cep135 (J). (I) $N = 764$ cells (Ctrl) and $N = 572$ (Cep120-KO). (J) $N = 515$ cells (Ctrl) and $N = 620$ (Cep120-KO). Data information: $N \geq 5$ mice per group. A two-tailed unpaired $t$ test was used for analyses and $p$ value denoted as follows: ***$p < 0.001$, ****$p < 0.0001$. The vertical segments in the box plots show the first quartile, median, and third quartile. The whiskers on both ends represents the maximum and minimum for each dataset analyzed.

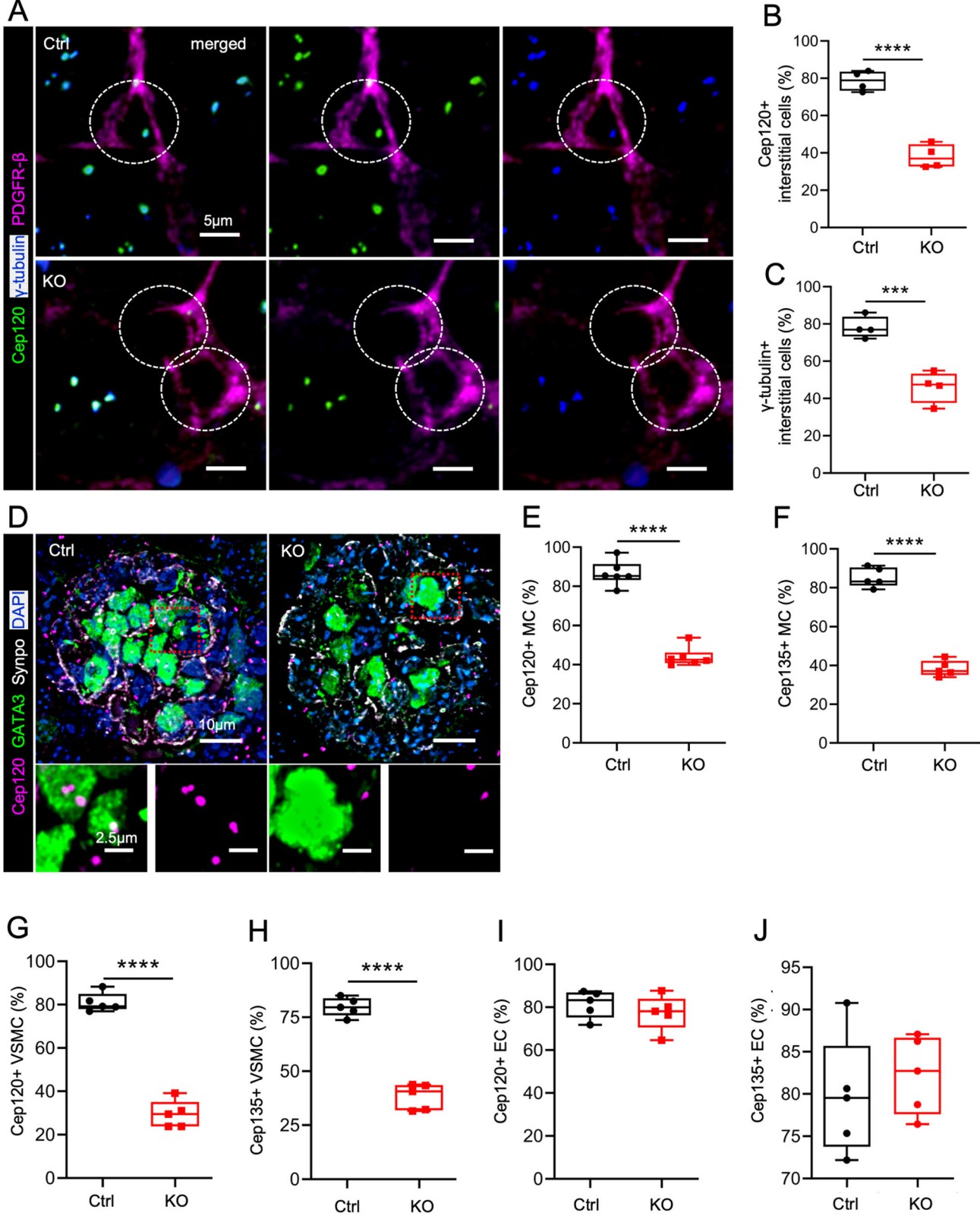

## A

**P15**

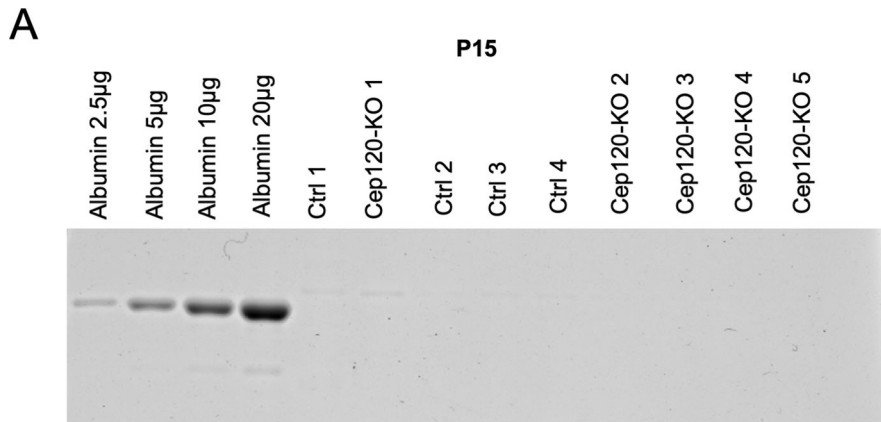

## B

**P150**

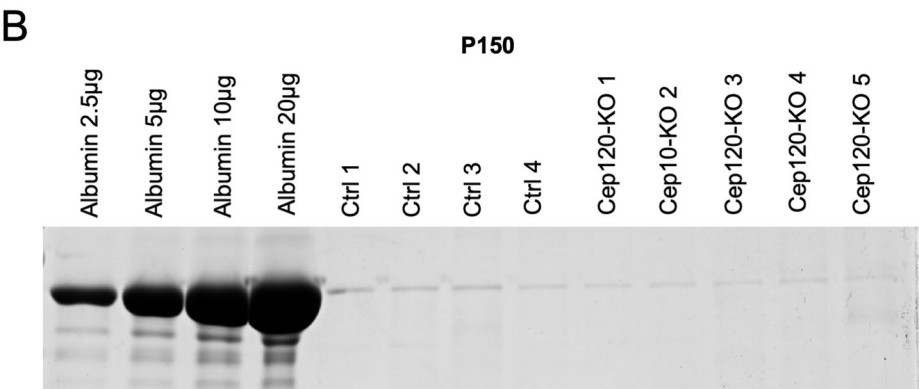

## C

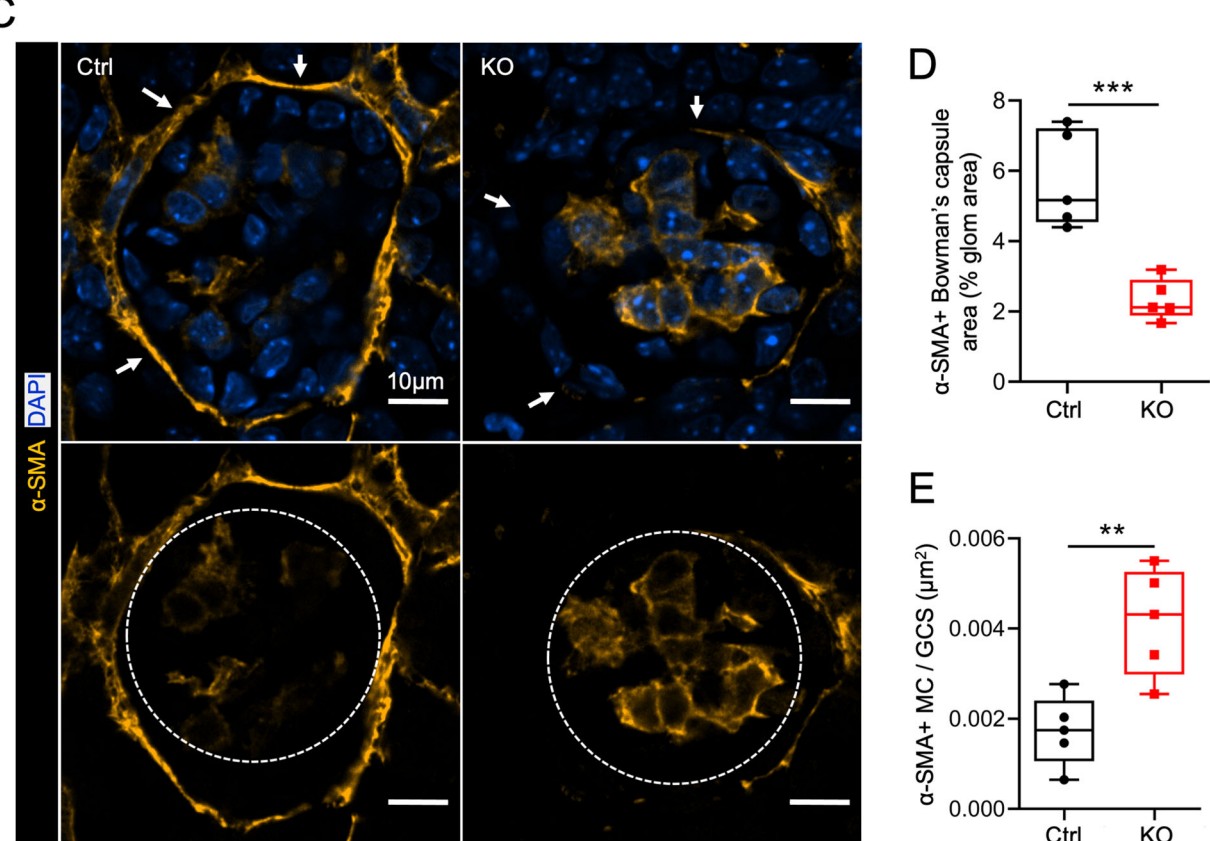

**D**

α-SMA+ Bowman's capsule area (% glom area)

***

Ctrl    KO

**E**

α-SMA+ MC / GCS (µm²)

**

Ctrl    KO

◀ **Figure EV2. Cep120 and centrosome loss causes defects in FoxD1-derived pericytes surrounding the Bowman's capsule and glomerular mesangial cells.**

(A, B) Coomassie gel of urine samples from control and Cep120-KO mice at (A) P15 and (B) P150. Bovine serum albumin (BSA) standards (2.5, 5, 10 and 20 μg) were also run on both gels. (C) Immunofluorescence staining of P15 kidney sections with anti α-smooth muscle actin (α-SMA) antibodies to mark pericytes surrounding Bowman's capsule (marked with white arrows; upper panel) and mesangial cells (lower panel). (D) Quantification of α-SMA-positive Bowman's capsule area expressed as percentage of total glomerular area. $N = 65$ glomeruli (Ctrl) and $N = 58$ (Cep120-KO). (E) Quantification of α-SMA-positive mesangial cell density per glomerular cross-sectional area. $N = 303$ cells (Ctrl) and $N = 613$ (Cep120-KO). Data information: $N = 5$ mice per group. A two-tailed unpaired $t$ test was used for analyses and $p$ value denoted as follows: **$p < 0.01$, ***$p < 0.001$. The vertical segments in the box plots show the first quartile, median, and third quartile. The whiskers on both ends represents the maximum and minimum for each dataset analyzed.

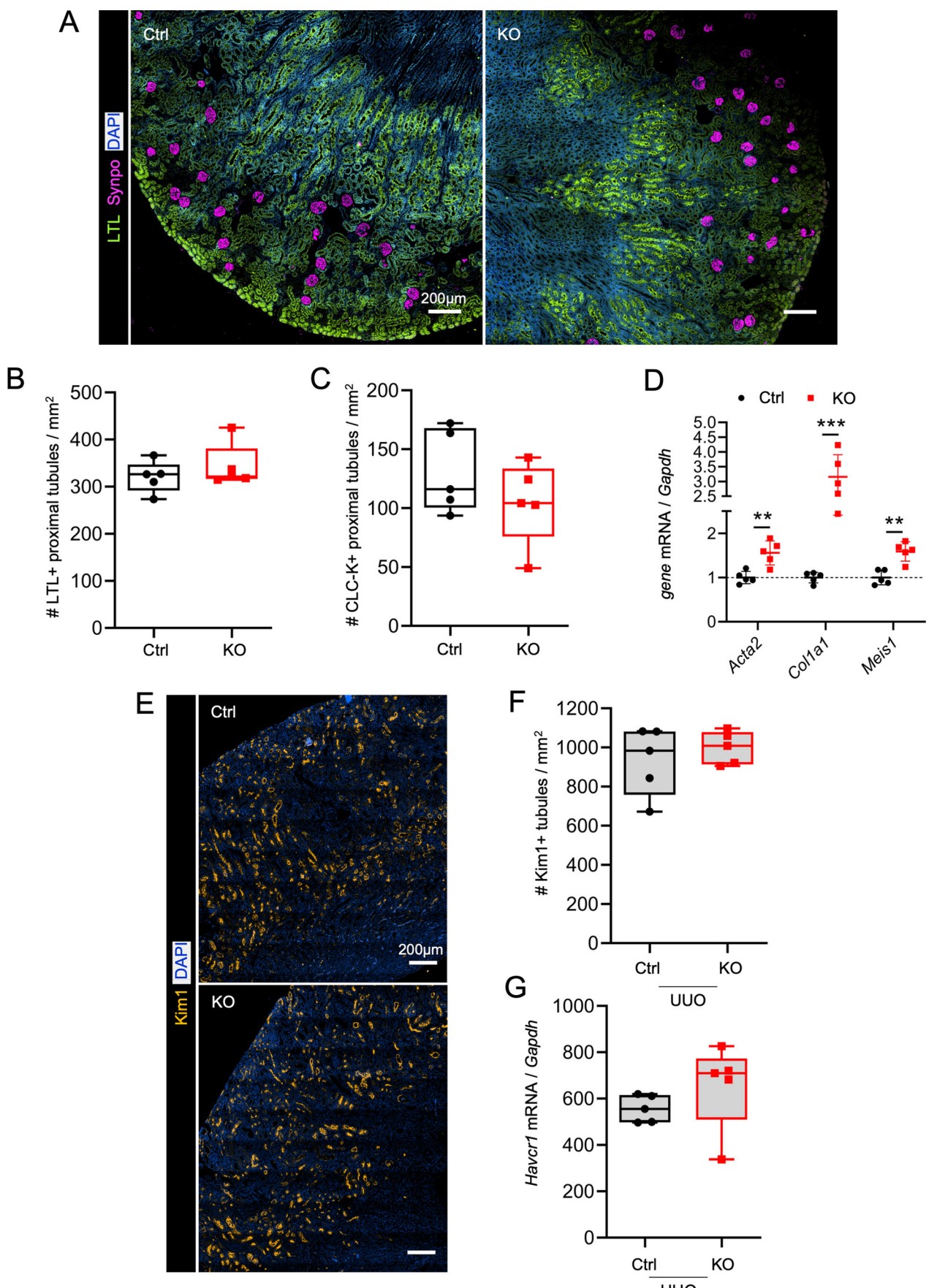

**Figure EV3. Analysis of nephron number and maturation, as well as transcriptional expression of profibrotic factors in long-term Cep120-KO survivors.**

(A) Immunofluorescence staining of P150 control and Cep120-KO kidney sections with antibodies to mark glomeruli (synaptopodin) and proximal tubules (LTL). (B, C) Quantification of LTL-positive proximal tubules and CLC-K-positive distal tubules per unit area. (B) $N = 1457$ tubules (Ctrl) and N = 1557 (Cep120-KO). (C) $N = 598$ tubules (Ctrl) and $N = 477$ (Cep120-KO). (D) qPCR-based quantification of the relative change in gene expression levels of Acta2, Col1a1 and Meis1 in control and Cep120-KO kidneys at P150. (E) Immunofluorescence staining of P60 kidney sections with antibodies to kidney injury marker 1 (Kim1) in control and Cep120-KO mice following UUO injury. (F) Quantification of Kim1-positive tubule number per unit area. $N = 938$ tubules (Ctrl) and $N = 991$ (Cep120-KO). (G) qPCR-based quantification of the relative change in gene expression levels of Havcr1 in control and Cep120-KO kidneys after 7 days of injury (UUO) and without injury (sham), expressed as fold change of sham-control. UUO was performed in 2 months old mice. Data information: $N = 5$ mice per group. A two-tailed unpaired $t$ test was used for analyses, and p-value denoted as follows: $**p < 0.01$, $***p < 0.001$. The vertical segments in the box plots show the first quartile, median, and third quartile. The whiskers on both ends represents the maximum and minimum for each dataset analyzed.

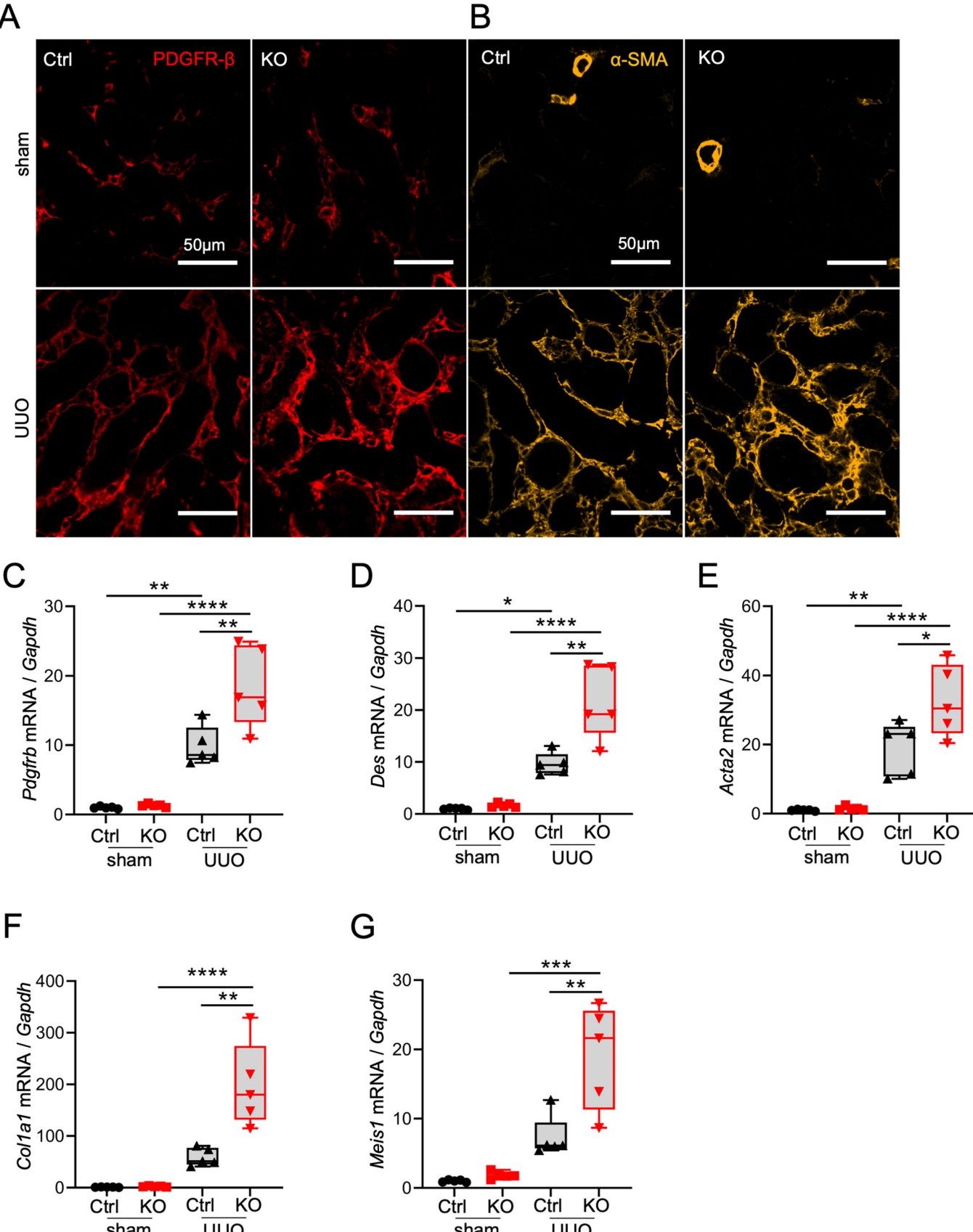

◄ **Figure EV4. Defective centrosome biogenesis in the stroma accelerates injury-induced fibrosis.**

(A, B) Immunofluorescence staining of P60 kidney sections with antibodies to mark (A) pericytes/fibroblasts (PDGFR-β) and (B) myofibroblasts (α-SMA) in sham (upper panel) and UUO kidneys (lower panel). (C–G) qPCR-based quantification of the relative change in gene expression levels of (C) Pdgfrb, (D) desmin, (E) Acta2, (F) Col1a1, and (G) Meis1 in control and Cep120-KO kidneys after 7 days of injury (UUO) and without (sham), expressed as fold change relative to sham-control. Data information: $N = 5$ mice per group. A one-way ANOVA test followed by multiple-group comparison analysis with Tukey correction was used for analyses, and p-value denoted as follows: $*p < 0.05$, $**p < 0.01$, $***p < 0.001$, $****p < 0.0001$. The vertical segments in the box plots show the first quartile, median, and third quartile. The whiskers on both ends represents the maximum and minimum for each dataset analyzed.

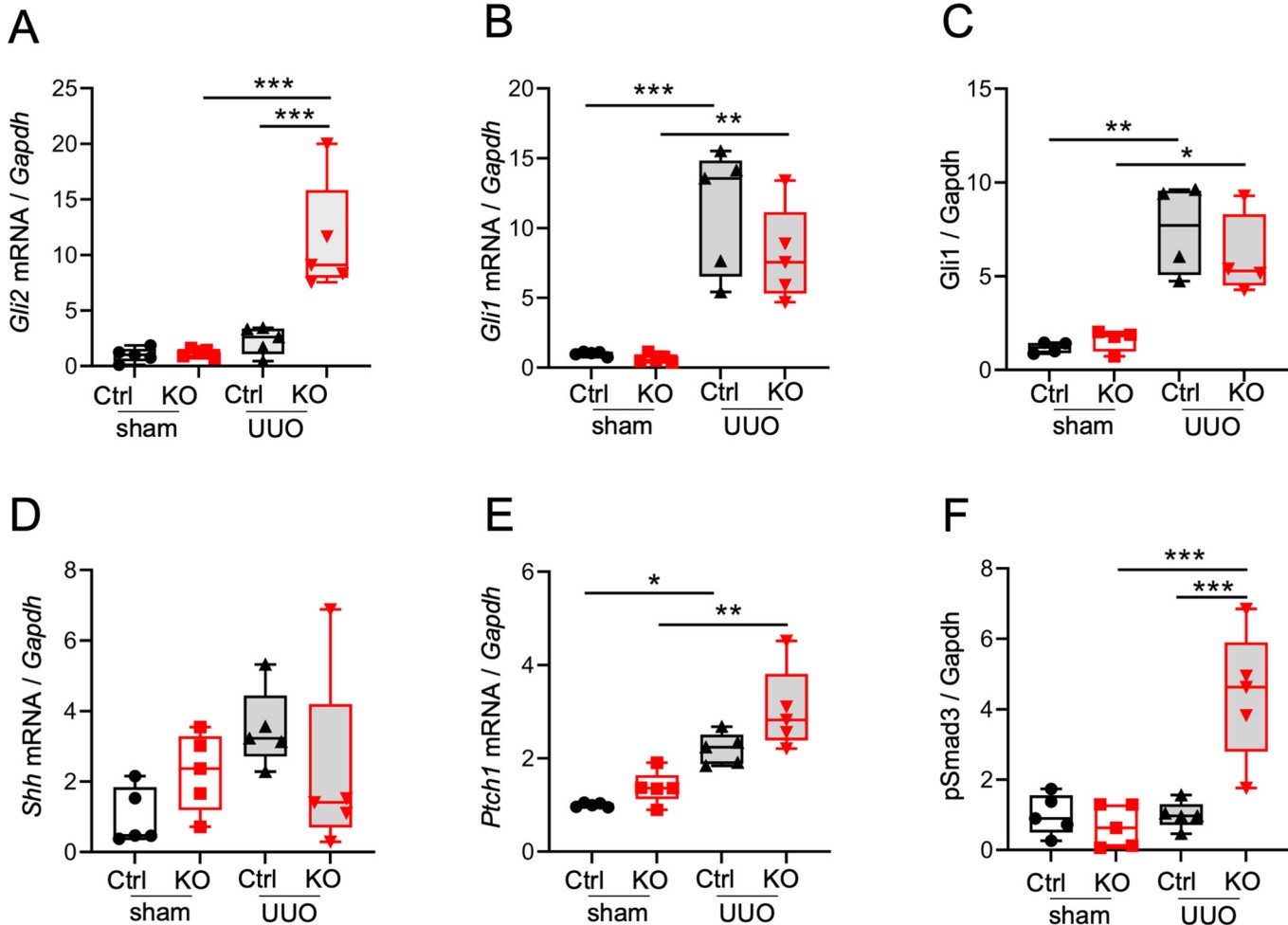

**Figure EV5. Quantification of signaling factors implicated in injury-induced fibrosis.**

(A, B) qPCR-based quantification of the relative change in gene expression levels of (A) Gli2 and (B) Gli1 in control and Cep120-KO kidneys after 7 days of injury (UUO) and without injury (sham), expressed as fold change relative to sham-control. (C) Quantification of Gli1 protein levels normalized to Gapdh, expressed as fold change relative to sham-control. (D, E) qPCR-based quantification of the change in gene expression levels of (D) Shh and (E) Ptch1 in control and Cep120-KO kidneys after 7 days of injury (UUO) and without (sham), expressed as fold change relative to sham-control. (F) Quantification of pSmad3 protein levels normalized to Gapdh, expressed as fold change of sham-control. Data information: $N = 5$ mice per group. A one-way ANOVA test followed by multiple-group comparison analysis with Tukey correction was used for analyses, and $p$ value denoted as follows: $*p < 0.05$, $**p < 0.01$, $***p < 0.001$. The vertical segments in the box plots show the first quartile, median, and third quartile. The whiskers on both ends represents the maximum and minimum for each dataset analyzed.

