## [Peer Review File · EMBO Reports]

Cep120 is essential for kidney stromal progenitor cell growth and differentiation.

Ewa Langner, Tao Cheng, Eirini Kefaloyianni, Charles Gluck, Baolin Wang, and Moe Mahjoub

DOI: [10.15252/embr.202357774](https://doi.org/10.15252/embr.202357774)

Corresponding author(s): Moe Mahjoub (mmahjoub@wustl.edu)

Review Timeline:

Submission Date:	7th Jul 23
Editorial Decision:	14th Sep 23
Revision Received:	14th Oct 23
Editorial Decision:	15th Nov 23
Revision Received:	15th Nov 23
Accepted:	22nd Nov 23

Editor: Deniz Senyilmaz Tiebe

Transaction Report:

Dear Moe,

Thank you for the submission of your research manuscript to our journal, which was now seen by three referees, whose reports are copied below.

My apologies for this unusual delay in getting back to you. It took longer than anticipated to receive the full set of referee reports.

The referees express interest in the proposed role of Cep120/centrosomal biogenesis in kidney stromal progenitor cell growth and differentiation. However, they also raise significant concerns that need to be addressed to consider publication here.

Of note, we notice your preprint (Cheng et al., 2023), which investigates the role of Cep120 in the metanephric mesenchyme (MM) and ureteric bud (UB) progenitor cells, and thus relevant for the current submission (<https://www.biorxiv.org/content/10.1101/2023.04.04.535568v1>). This was also pointed out by referees #1 and #3. Please note that per our editorial policy that any related work under consideration, review, revision or accepted for publication elsewhere should have accompanied the initial submission if they are relevant to its scientific assessment. Therefore, please update us on the status of the preprint by responding to this email. If it is already in press, please cite it as a paper in the revised manuscript. Moreover, please discuss Cheng et al. 2023 in more detail in the revised manuscript by making it clear that it is from your group, as also requested by referee #3.

Given the positive recommendations of the referees, we would like to invite you to submit a revised manuscript. Please revise your manuscript with the understanding that the referee concerns (as in their reports) must be fully addressed and their suggestions taken on board. Please address all referee concerns in a complete point-by-point response. Acceptance of the manuscript will depend on a positive outcome of a second round of review. It is EMBO reports policy to allow a single round of major experimental revision only and acceptance or rejection of the manuscript will therefore depend on the completeness of your responses included in the next, final version of the manuscript.

We realize that it is difficult to revise to a specific deadline. In the interest of protecting the conceptual advance provided by the work, we recommend a revision within 3 months. Please discuss the revision progress ahead of this time with me if you require more time to complete the revisions, or if you have questions or comments regarding the revision (also by video chat).

1. A data availability section providing access to data deposited in public databases is missing (where applicable).
2. Your manuscript contains statistics and error bars based on $n=2$. Please use scatter plots in these cases.

You can submit the revision either as a Scientific Report or as a Research Article. For Scientific Reports, the revised manuscript can contain up to 5 main figures and 5 Expanded View figures, and it should not exceed 27000 characters. If the revision leads to a manuscript with more than 5 main figures it will be published as a Research Article. In this case the Results and Discussion section should be separate. If a Scientific Report is submitted, these sections have to be combined. This will help to shorten the manuscript text by eliminating some redundancy that is inevitable when discussing the same experiments twice. In either case, all materials and methods should be included in the main manuscript file.

2) individual production quality figure files as .eps, .tif, .jpg (one file per figure). See https://wol-prod-cdn.literatumonline.com/pb-assets/embosite/EMBOPress_Figure_Guidelines_061115-1561436025777.pdf for more info on how to prepare your figures.

- Additional Tables/Datasets should be labeled and referred to as Table EV1, Dataset EV1, etc. Legends have to be provided in

a separate tab in case of .xls files. Alternatively, the legend can be supplied as a separate text file (README) and zipped together with the Table/Dataset file.

4) a .docx formatted letter INCLUDING the reviewers' reports and your detailed point-by-point responses to their comments. As part of the EMBO publication's Transparent Editorial Process, EMBO reports publishes online a Review Process File (RPF) to accompany accepted manuscripts. This File will be published in conjunction with your paper and will include the referee reports, your point-by-point response and all pertinent correspondence relating to the manuscript.

<https://www.embopress.org/page/journal/14693178/authorguide#transparentprocess>

5) a complete author checklist, which you can download from our author guidelines

<https://www.embopress.org/page/journal/14693178/authorguide>. Please insert information in the checklist that is also reflected in the manuscript. The completed author checklist will also be part of the RPF.

6) Please note that all corresponding authors are required to supply an ORCID ID for their name upon submission of a revised manuscript (<<https://orcid.org/>>). Please find instructions on how to link your ORCID ID to your account in our manuscript tracking system in our Author guidelines

<<https://www.embopress.org/page/journal/14693178/authorguide#authorshippinguidelines>>

7) Before submitting your revision, primary datasets produced in this study need to be deposited in an appropriate public database (see <https://www.embopress.org/page/journal/14693178/authorguide#datadeposition>). Please remember to provide a reviewer password if the datasets are not yet public. The accession numbers and database should be listed in a formal "Data Availability" section placed after Materials & Method (see also

<https://www.embopress.org/page/journal/14693178/authorguide#datadeposition>). Please note that the Data Availability Section is restricted to new primary data that are part of this study. * Note - All links should resolve to a page where the data can be accessed. *

Additional information on source data and instruction on how to label the files are available:

<https://www.embopress.org/page/journal/14693178/authorguide#sourcedata>

9) Our journal encourages inclusion of *data citations in the reference list* to directly cite datasets that were re-used and obtained from public databases. Data citations in the article text are distinct from normal bibliographical citations and should directly link to the database records from which the data can be accessed. In the main text, data citations are formatted as follows: "Data ref: Smith et al, 2001" or "Data ref: NCBI Sequence Read Archive PRJNA342805, 2017". In the Reference list, data citations must be labeled with "[DATASET]". A data reference must provide the database name, accession number/identifiers and a resolvable link to the landing page from which the data can be accessed at the end of the reference. Further instructions are available at <http://www.embopress.org/page/journal/14693178/authorguide#referencesformat>

10) Regarding data quantification (see Figure Legends:

<https://www.embopress.org/page/journal/14693178/authorguide#figureformat>)

- the name of the statistical test used to generate error bars and P values,

- the number (n) of independent experiments (please specify technical or biological replicates) underlying each data point,

- the nature of the bars and error bars (s.d., s.e.m.),

- If the data are obtained from n Program fragment delivered error ``Can't locate object method "less" via package "than" (perhaps you forgot to load "than"?) at //ejpvfs23/sites23b/embor_www/letters/embor_decision_revise_and_review.txt line 56.' 2, use scatter blots showing the individual data points.

Discussion of statistical methodology can be reported in the materials and methods section, but figure legends should contain a

basic description of n, P and the test applied.

12) Please also note our reference format:

I look forward to seeing a revised version of your manuscript when it is ready. Please let me know if you have questions or comments regarding the revision.

Kind regards,

Deniz

Deniz Senyilmaz Tiebe, PhD
Editor
EMBO Reports

Referee #1:

The manuscript by Langner et al. discusses the role of the centrosomal protein Cep120 in kidney stromal progenitor cells during kidney development and growth, specifically in the stromal mesenchyme. At the cellular level, the loss of Cep120 is crucial for centrosome duplication and leads to centrosome loss (CL). Pathogenetic variants of CEP120 have been identified to cause ciliopathies (JS and JATD) characterized by severe developmental defects of the kidney and renal fibrosis/cysts. A preprint of the same lab describes the loss of CEP120 in the cap mesenchyme and the ureteric bud epithelia, both resulting in massive cystic kidney disease, consistent with the important role of cilia/centrosomes in kidney tubular epithelial cells.

In the at hand study, the authors employed a conditional knockout in the stromal mesenchyme using FOxD1 cre, which impacted the numbers of pericytes, fibroblasts, and mesangial cells. Mechanistically, consistent with the centrosomal functions of Cep120, the authors observed alterations in cell cycle progression and increased p53 and caspase 3 activity, leading to apoptosis. Additionally, the authors noted a delay in nephron maturation. This indirect regulation through stromal cells appears to be mediated by Wnt and Hedgehog (HH) signaling. In the injury model (unilateral ureteral obstruction - UOO), the loss of Cep120 did not influence tubular injury; however, markers associated with fibroblasts and fibrosis showed upregulation, along with altered TGF-beta signaling.

In my perspective, this study addresses a remarkably captivating topic and holds significant importance and novelty. Particularly, it highlights an often overlooked area within ciliopathy research-the interstitial cell lineages of the kidney. Traditionally, the focus has predominantly been on tubular epithelial cells as the origin of cyst formation. It's indeed overdue to shift this focus towards a more comprehensive understanding of kidney pathology. Given this context, I find the present paper highly commendable.

The absence of spontaneous cyst formation and fibrosis, as observed by the authors, holds profound implications for the entire ciliopathy field, particularly within nephrology. Moreover, the paper has a high level of novelty. The quality of the imaging data is truly impressive. The paper is exceptionally well-written and the data well presented.

Hence, I find myself somewhat conflicted about any potential delay in publishing this exceptional study, which I feel is exceptionally well-suited for the Journal EMBO Reports.

However, there are a few minor concerns I believe the authors should promptly address:

1. The substantial decrease in mesangial cells, as indicated by the anti-desmin immunofluorescence signal, is indeed quite remarkable. Even though the synpo signal appears to remain largely unaffected, it would be beneficial to ascertain whether the reduction in mesangial cells has any impact on the integrity of the filtration barrier. Do these mice develop proteinuria?
2. The presence of clustered cleaved caspase 3-positive cells, as depicted in Figure 4I, raises an interesting question. Is this

clustering observed solely in this specific image, or does it represent a broader finding? Exploring the underlying reasons for such localized cell death could provide valuable insights. Can you speculate what factors might contribute to this focal pattern of cell death?

3. This certainly goes far beyond the study, but are there speculations about what the animals are dying from? Are they dying from the consequences of CKD, such as cardiovascular events?

4. The authors should not forget to mention the use of Biorender for the graphical abstract / Fig. 1A (if this is not redone by EMBO press).

5. Please replace red color in IF pictures (e.g., Fig 4G/I) by magenta (with respect to color blind people).

Referee #2:

Summary

In this manuscript, Lengner et al. report that aberrant centrosome biogenesis in the mouse embryo caused reduced abundance of several kidney stromal cell populations, leading to the development of small atrophied kidneys in adult mice. The findings reported here are convincing, using independent lines of experimental evidence and done with the appropriate controls. The manuscript is easy to read, scientifically correct without being fastidious and well put together, in the sense that there is a clear understanding of the main message to convey and what is accessory data.

The consequences of centrosome loss (CL) have been studied extensively in a variety of animal models and tissues, and my overall assessment is that this manuscript is largely descriptive- listing biological functions that do and do not change in the developing and adult kidney when CL is induced in a subset of cells in the developing kidney. Moreover, some effects are cell autonomous, while others are not, making it hard to infer causal relationships. The main significance of this report is the thorough description of the defects that arise when CL is induced kidney progenitors, which will be of interest to those in the centriole/cilia fields.

Below I list a number of points that the authors should consider.

Major Comments

1. The authors should state how many KO cells were counted in each experiment. For example, the Fig.1 legend states that the quantifications and statistical analysis (in Figs. 1D and 1F) correspond to 4 mice per group but it doesn't state how many cells were quantified per mouse/per tissue section. This is true for most figures shown in the article.

2. Fig. 4C shows kidneys from pregnant females injected with EdU to label replicating DNA in S-phase cells and co-stained with pHH3 to mark G2/M phases of the cell cycle. The authors conclude that there was a marked increase in Aldh1a2/pHH3 cells which are SM cells in M phase. I found this figure challenging to interpret as it's difficult to unambiguously discern the boundaries of the Aldh1a2 cells (green staining). For example, there are two cells in the control panel expressing pHH3 (pink) but it's not clear whether these are Aldh1a2 cells. The authors should describe how many Aldh1a2 cells were counted in total, the method used to count cells and the precise criteria used to score cells as Aldh1a2 positive, as there seems a lot of variability. The same is true for scoring with several other of the cell identity markers.

3. Figs. 4G and 4I shows increased expression of P53 and CC3, respectively, in KO cells. This is a clear result, both visually and quantitatively, although here the authors are quantifying cells/mm² rather than % of P53 cells in the Meis1 population, similarly to what was done above. Was there a specific reason for this change in methodology? In any case, the percentage of cells undergoing apoptosis still seems very low. I'd be curious to know if cells are dying quickly at some point during development and then we're seeing the aftermath here and in the adult kidneys or are cells dying slowly and there's enough time for the tissue to regenerate? It's not immediately apparent if apoptosis is a significant factor that leads to the range of atrophies seen in the adult kidney (Fig. 2C). Imaging at various stages of development stages of development would be ideal, but I understand that this might not be feasible as it would require many mice.

Minor Comments

1. Fig.1C shows that in the CEP120 KO mice embryos there were 40% of SM cells expressing CEP120 vs 80% in the control embryos. Why is there a population of cells in the wt not expressing CEP120? Is this real or an artifact of the IF staining? Furthermore, why is there still a sizeable population of cells in the KO expressing CEP120? I couldn't find an explanation in the text. Ditto for gamma-tubulin. These are significant, because it indicates that centrosome biogenesis was not completely abolished.

2. In the main text it is stated that "Cep120 expression and centrosome assembly were unaffected in cells not derived from FoxD1 progenitors (e.g., tubular epithelia and endothelial cells; Fig S11)". This Fig. shows CEP120 quantification but I couldn't find an associated quantification of gamma-tubulin to discard the possibility of defects in centrosome biogenesis, as done above.
3. Figs. 2F and 2G it is stated that there was an increase in creatinine and BUN levels in serum of CEP120 KO adult mice compared to the control mice but that this was within physiological range. Please state the physiological range and what would be considered abnormal (for the benefit of a non-specialized audience), for example P15 and P150 have BUN levels up by roughly 50% compared to the control.
4. Fig.3 shows kidneys of P15 mice stained with pericyte, interstitial fibroblast markers and glomerular mesangial cells. There was a significant decrease in the populations of these cells in CEP120 KO mice compared to control, but the percentage decrease is missing.
5. Fig.6 shows a comparison between the kidneys of CEP120 KO vs wt mice subjected to UUO, 7 days post procedure. There was an elevated expression of pericyte and fibroblast markers (desmin, PDGFR-beta, Meis1) at RNA and protein level in KO kidneys + upregulation of fibrosis and ECM deposition markers fibronectin, α -SMA, collagen). In the sham control, why is fibronectin not elevated in the CEP120 KO mice compared to control, given that the authors showed elevated ECM deposition in KO kidneys above?

Referee #3:

In this manuscript entitled « Cep120 is essential for kidney stromal progenitor cell growth and differentiation » Langner and colleagues analyze the consequences of centrosome dysfunction on renal progenitor cell physiology. To perform this study, the authors prevent the expression of the ciliopathic gene Cep120 in the stromal mesenchyme, using a mouse model harboring a floxed allele of Cep120 crossed with a FoxD1-Cre strain, referred as Cep120-KO. Due to extra renal phenotypes, some mice die at P15, while others lasted up to 5 months. At P15, kidneys appear hypoplastic with medullary atrophy, enlarged collecting duct tubules and delayed nephron maturation. These phenotypes were due to delayed mitosis, activation of the mitotic surveillance pathway and changes in Wnt and Hedgehog signaling. However, kidney function appear normal. After renal injury using unilateral ureteral obstruction, renal fibrosis is observed due to enhanced TGF- β /Smad3-Gli2 signaling.

The article is well-written, the data are mostly convincing nevertheless some questions remain.

As a general comment, since EMBO Reports readers are often not familiar with kidney, is it possible to help the reader by giving more details in the figure legend?

1) The daughter centriole ciliopathic protein CEP120 is required for centriole duplication. In another hand, it regulates microtubule centriole length with CPAP. In the paper, the authors claimed that the conditional deletion of Cep120 gene lead to a reduced number of cells expressing CEP120 with a reduced number of centrosome, interpreted as centrosome loss due to the involvement of CEP120 in centriole duplication.

In my opinion, the effective loss of centrosome is not demonstrated. γ -tubulin staining is used to detect the centrosome. But a single centrosome marker is not enough to demonstrate the effective loss of centrosome. CEP120 has been shown to be involved in the elongation of the microtubule centriolar wall and its depletion in cultured cells leads to small centrioles. Therefore, can the authors use other markers as for example CP110, which is supposed to remain on the centriole after CEP120 depletion (Tsai et al, 2019). Another possibility could be to perform FIB-SEM to demonstrate the absence of centriolar structure.

2) Concerning the mitosis phenotype, can the authors present another field more convincing (the magnified cell is OK, but the other one indicated by an arrow is strange since it is both Edu positive and pHH3 positive, why?) and a second field in sup data?

3) Fig 4G, CEP120-KO: the authors show merged image, is it possible to put single channels to highlight better the p53 positive cells?

4) According to Fig 4G CEP120-KO, p53 positive cells are scattered in the tissue, why only one cluster of apoptotic cells is observed (Fig 4I)?

Minor point:

In the discussion part, the authors wrote "In a concurrent study". The study has been done in the same lab. Can the authors change the sentence?

Referee #1:

In my perspective, this study addresses a remarkably captivating topic and holds significant importance and novelty. Particularly, it highlights an often-overlooked area within ciliopathy research-the interstitial cell lineages of the kidney. Traditionally, the focus has predominantly been on tubular epithelial cells as the origin of cyst formation. It's indeed overdue to shift this focus towards a more comprehensive understanding of kidney pathology. Given this context, I find the present paper highly commendable.

The absence of spontaneous cyst formation and fibrosis, as observed by the authors, holds profound implications for the entire ciliopathy field, particularly within nephrology. Moreover, the paper has a high level of novelty. The quality of the imaging data is truly impressive. The paper is exceptionally well-written and the data well presented.

Hence, I find myself somewhat conflicted about any potential delay in publishing this exceptional study, which I feel is exceptionally well-suited for the Journal EMBO Reports.

- We would like to thank the reviewer for the extremely positive assessment of our work. We agree that this is an important and under-studied topic.

However, there are a few minor concerns I believe the authors should promptly address:

*1. The substantial decrease in mesangial cells, as indicated by the anti-desmin immunofluorescence signal, is indeed quite remarkable. Even though the synpo signal appears to remain largely unaffected, it would be beneficial to ascertain whether the reduction in mesangial cells has any impact on the integrity of the filtration barrier. Do these mice develop **proteinuria**?*

- We analyzed urine isolated from Cep120-KO mice at both P15 and P150, and it does not appear to show any substantial proteinuria. Quantification of albumin/creatinine ratio showed no difference between control and Cep120-KO mice. These new data are presented as Figure 2H and EV2A and B. This finding is also consistent with the negligible change in blood urea nitrogen levels (Fig. 2G) in the Cep120-KO mice.

2. The presence of clustered cleaved caspase 3-positive cells, as depicted in Figure 4I, raises an interesting question. Is this clustering observed solely in this specific image, or does it represent a broader finding? Exploring the underlying reasons for such localized cell death could provide valuable insights. Can you speculate what factors might contribute to this focal pattern of cell death?

- Although we do notice areas of single cleaved caspase 3-positive cells, we predominantly observed the presence of clustered apoptotic cells throughout

multiple regions of each Cep120-KO kidney. Therefore, the image is truly representative of what we see in our immunostaining experiments. There are at least two possible explanations for this phenomenon: 1) First, the clusters of cleaved caspase 3-positive cells may be due to enrichment in areas where Cep120-ablated cells (and their derived daughter cells) are present. This makes sense as the loss of centrosomes would occur during the cell division phase, and the daughter cells would then be nearby. 2) Another explanation of this phenomena might be a study published by Boivin and Bridgewater (AJP-Renal 2018, which we cite in the manuscript) describing clusters of apoptotic cells specifically in the corticomedullary junction of kidneys, following stromal β -catenin deletion. As our stromal deletion of centrosomes leads to deregulation of many of the same signaling pathways (as shown in the Fig. 5G), this could result in the localized cell death patterns that we observed.

3. This certainly goes far beyond the study, but are there speculations about what the animals are dying from? Are they dying from the consequences of CKD, such as cardiovascular events?

- We do not believe that these animals are dying due to CKD, as their kidney filtration functions are mostly intact. We believe the mice are dying due to extra-renal manifestations. As we mention in the Results and Discussion sections: "(...) the early lethality observed in Cep120-KO mice is likely due to extrarenal phenotypes, as FoxD1 is also expressed in the anterior hypothalamus, retinal ganglion, ventral diencephalon and lung pericytes (Carreres et al., 2011, Newman et al., 2018, Herrera et al., 2004, Hung et al., 2013). This lethality has been previously reported when ablating genes using FoxD1-Cre-expressing mice (Nie and Arend, 2017, Karolak et al., 2018)." Therefore, we think that the short-term Cep120-KO survivors are likely dying due to brain malformations (hydrocephalus) as they usually showed a dome-shaped heads and short snouts. We did not investigate this phenotype in detail as it is well known that centrosome loss in the brain leads to brain abnormalities, and our focus was on the kidney interstitial tissue.

4. The authors should not forget to mention the use of Biorender for the graphical abstract / Fig. 1A (if this is not redone by EMBO press).

- Yes! We thank the reviewer for pointing out this oversight. We have added this information to the Acknowledgment section.

5. Please replace red color in IF pictures (e.g., Fig 4G/I) by magenta (with respect to color blind people).

- We have adjusted all of the figures that contained both red and green, as requested.

Referee #2:

The consequences of centrosome loss (CL) have been studied extensively in a variety of animal models and tissues, and my overall assessment is that this manuscript is largely descriptive- listing biological functions that do and do not change in the developing and adult kidney when CL is induced in a subset of cells in the developing kidney. Moreover, some effects are cell autonomous, while others are not, making it hard to infer causal relationships. The main significance of this report is the thorough description of the defects that arise when CL is induced kidney progenitors, which will be of interest to those in the centriole/cilia fields.

- We thank the reviewer for highlighting the significance of this report for the centriole-cilia field.

Major Comments

1. The authors should state how many cells were counted in each experiment. For example, the Fig.1 legend states that the quantifications and statistical analysis (in Figs. 1D and 1F) correspond to 4 mice per group but it doesn't state how many cells were quantified per mouse/per tissue section. This is true for most figures shown in the article.

- We thank the reviewer for pointing out this oversight. We have included all numerical information collected for each experiment in the appropriate figure legends.

2. Fig. 4C shows kidneys from pregnant females injected with EdU to label replicating DNA in S-phase cells and co-stained with pHH3 to mark G2/M phases of the cell cycle. The authors conclude that there was a marked increase in Aldh1a2/phh3 cells which are SM cells in M phase. I found this figure challenging to interpret as it's difficult to unambiguously discern the boundaries of the Aldh1a2 cells (green staining). For example, there are two cells in the control panel expressing pHH3 (pink) but it's not clear whether these are Aldh1a2 cells. The authors should describe how many Aldh1a2 cells were counted in total, the method used to count cells and the precise criteria used to score cells as Aldh1a2 positive, as there seems a lot of variability. The same is true for scoring with several other of the cell identity markers.

- For presentation purposes, the images presented in Fig 4C are a maximum-intensity projection. This makes the image a bit more “busy” looking, and the cell boundaries harder to discern. However, our quantifications were performed using the raw data (i.e. images that contained the Z-sections). This way, we could confirm that cells were Aldh1a2-positive, while also carefully scoring for the presence or absence of pHH3. We have added this detail to the Methods section. Moreover, we note that >1000 cells were scored from 5 individual kidneys for these analyses, which helps overcome the variability, and ensure rigor of the result.

- We have also added the total number of cells counted for each group to the figure legend, as requested.

3. Figs. 4G and 4I shows increased expression of P53 and CC3, respectively, in KO cells. This is a clear result, both visually and quantitatively, although here the authors are quantifying cells/mm² rather than % of P53 cells in the Meis1 population, similarly to what was done above. Was there a specific reason for this change in methodology?

- In fact, all of the quantifications performed in the study are represented per unit area, as is standard in the nephrology field. Moreover, this is done specifically because the kidney size in the Cep120-KO mice is smaller, which requires us to factor in this difference. The one exception to this rule is the cell cycle analyses in Fig4 D-F. In that case, we were examining at the same population of cells (Aldh1a2+) but needed to represent which fraction of those cells are in each cell cycle stage. Hence, the need to use “% of cells” instead of “cells per unit area” as was done throughout the study.

In any case, the percentage of cells undergoing apoptosis still seems very low. I'd be curious to know if cells are dying quickly at some point during development and then we're seeing the aftermath here and in the adult kidneys or are cells dying slowly and there's enough time for the tissue to regenerate? It's not immediately apparent if apoptosis is a significant factor that leads to the range of atrophies seen in the adult kidney (Fig. 2C). Imaging at various stages of development stages of development would be ideal, but I understand that this might not be feasible as it would require many mice.

- We note that the quantification of cell death shown in Fig 4J was in fact done during the developmental stage (E15.5). Therefore, we are not necessarily observing the aftermath of large-scale cell death. Note that there is a roughly 3-fold increase in apoptosis upon Cep120 loss. We believe that this level of cell death adds up over the time of kidney development, and is consistent with the fractional decrease in the derived cell populations (e.g. Fig 3B-F). Since we did not isolate kidneys at earlier developmental stages, unfortunately this analysis was not feasible here.

Minor Comments

1. Fig.1C shows that in the CEP120 KO mice embryos there were 40% of SM cells expressing CEP120 vs 80% in the control embryos. Why is there a population of cells in the wt not expressing CEP120? Is this real or an artifact of the IF staining?

- This is an artifact of tissue sectioning. For tissue histology and immunofluorescence, we typically cut paraffin-embedded kidney samples at 7-

10 μ m thickness. As a result, the centrosomes of some cells are just not present in the section. This is a common issue with analysis of tissue sections.

Furthermore, why is there still a sizeable population of cells in the KO expressing CEP120? I couldn't find an explanation in the text. Ditto for gamma-tubulin. These are significant, because it indicates that centrosome biogenesis was not completely abolished.

- There are some cells still expressing both Cep120 and centrosomes due to partial KO of the gene in Aldh1a2-positive cells, and the fact that it is not deleted in neighboring cells (as expected). This is also partly due to the fact that Foxd1-Cre homozygous animals are embryonic lethal, thus we had to use the FoxD1-Cre mice as heterozygous animals in the background of Cep120F/F. Therefore, the deletion of Cep120 gene is not fully penetrant.

2. In the main text it is stated that "Cep120 expression and centrosome assembly were unaffected in cells not derived from FoxD1 progenitors (e.g., tubular epithelia and endothelial cells; Fig S1I)". This Fig. shows CEP120 quantification but I couldn't find an associated quantification of gamma-tubulin to discard the possibility of defects in centrosome biogenesis, as done above.

- We thank the reviewer for pointing out this missing piece of data. Endothelial cells were co-stained with a centriolar marker (Cep135, as done for other cell types) and the percentage quantified. As predicted based on Cep120 expression, endothelial cells contained the normal complement of centrosomes, confirming the specificity of Cep120 loss only in cells of FoxD1-derived lineage. This is now presented as new Fig EV1J.

3. Figs. 2F and 2G it is stated that there was an increase in creatinine and BUN levels in serum of CEP120 KO adult mice compared to the control mice but that this was within physiological range. Please state the physiological range and what would be considered abnormal (for the benefit of a non-specialized audience), for example P15 and P150 have BUN levels up by roughly 50% compared to the control.

- The typical physiological range for serum creatinine in mice is 0.06-0.25 mg/dL, and for BUN is 18-42 mg/dL. We have added this information to the relevant section in Results (page 7) where we describe the levels in our Cep120-KO mice.

4. Fig.3 shows kidneys of P15 mice stained with pericyte, interstitial fibroblast markers and glomerular mesangial cells. There was a significant decrease in the populations of these cells in CEP120 KO mice compared to control, but the percentage decrease is missing.

- We have added the percentage decrease for each one in the relevant Results section (page 8).

5. Fig.6 shows a comparison between the kidneys of CEP120 KO vs wt mice subjected to UUO, 7 days post procedure. There was an elevated expression of pericyte and fibroblast markers (desmin, PDGFR-beta, Meis1) at RNA and protein level in KO kidneys + upregulation of fibrosis and ECM deposition markers fibronectin, α -SMA, collagen). In the sham control, why is fibronectin not elevated in the CEP120 KO mice compared to control, given that the authors showed elevated ECM deposition in KO kidneys above?

- As we discuss in the manuscript, there is no spontaneous fibrosis in Cep120 KO mice at 2 months of age. Fibrosis only occurs following UUO injury at this stage, and is only observed in the injured kidney and not the sham kidney. Therefore, in the sham kidneys, there is no difference in ECM deposition between the control and Cep120-KO, and we would not expect to see an increase in fibronectin. The difference only becomes evident once you induce injury, which is the main take-home message of this figure/experiment.

Referee #3:

In this manuscript entitled « Cep120 is essential for kidney stromal progenitor cell growth and differentiation » Langner and colleagues analyze the consequences of centrosome dysfunction on renal progenitor cell physiology. To perform this study, the authors prevent the expression of the ciliopathic gene Cep120 in the stromal mesenchyme, using a mouse model harboring a floxed allele of Cep120 crossed with a FoxD1-Cre strain, referred as Cep120-KO. Due to extra renal phenotypes, some mice die at P15, while others lasted up to 5 months. At P15, kidneys appear hypoplastic with medullary atrophy, enlarged collecting duct tubules and delayed nephron maturation. These phenotypes were due to delayed mitosis, activation of the mitotic surveillance pathway and changes in Wnt and Hedgehog signaling. However, kidney function appear normal. After renal injury using unilateral ureteral obstruction, renal fibrosis is observed due to enhanced TGF- β /Smad3-Gli2 signaling.

The article is well-written, the data are mostly convincing nevertheless some questions remain.

- We would like to thank the reviewer for the positive assessment of our work.

1) The daughter centriole ciliopathic protein CEP120 is required for centriole duplication. In another hand, it regulates microtubule centriole length with CPAP. In the paper, the authors claimed that the conditional deletion of Cep120 gene lead to a reduced number of cells expressing CEP120 with a reduced number of centrosome, interpreted as centrosome loss due to the involvement of CEP120 in centriole duplication. In my opinion, the effective loss of centrosome is not demonstrated. α -tubulin staining is used to detect the centrosome. But a single centrosome marker is not enough to

demonstrate the effective loss of centrosome. CEP120 has been shown to be involved in the elongation of the microtubule centriolar wall and its depletion in cultured cells leads to small centrioles. Therefore, can the authors use other markers as for example CP110, which is supposed to remain on the centriole after CEP120 depletion (Tsai et al, 2019). Another possibility could be to perform FIB-SEM to demonstrate the absence of centriolar structure.

- In the original submission, we had verified centrosome loss using 3 separate markers: Cep120, Cep135, and γ -tubulin. We have now added 2 additional markers of centrioles (centrin) and centrosomes (Ninein). Kidneys of control and Cep120-KO mice at E15.5 were co-stained for centrin and Ninein, and centrosome abundance quantified using the dual markers. Consistent with the previous analyses, there is a significant decrease in centrosome number in the stromal population. Similar to Cep120, Cep135 and γ -tubulin loss, we observed a roughly 50% decrease in the presence of cells double marked with Ninein and centrin. This is now presented as new Fig 1G-H. We also tried immunostaining samples for Cp110 as suggested by the reviewer, however the staining did not work in the kidney sections. Nonetheless, all 5 centriole/centrosome markers used indicate that Cep120 ablation results in loss of centrioles and centrosomes in the stromal cells.

2) Concerning the mitosis phenotype, can the authors present another field more convincing (the magnified cell is OK, but the other one indicated by an arrow is strange since it is both Edu positive and pHH3 positive, why?) and a second field in sup data?

- We have replaced the image panel with the strange Edu-pHH3 positive cells with a new field from Cep120-KO kidneys (Fig 4C, right panel), which only highlights pHH3 in the Aldh1a2-positive cell.

3) Fig 4G, CEP120-KO:the authors show merged image, is it possible to put single channels to highlight better the p53 positive cells?

- We have included single channels of the immunofluorescent images from Fig 4G (separately for p53 and other markers) in the supplementary material (Appendix Fig S1).

4) According to Fig 4G CEP120-KO, p53 positive cells are scattered in the tissue, why only one cluster of apoptotic cells is observed (Fig 4I)?

- As mentioned above in response to the comment of reviewer 1 (point #2), although we do notice areas of single cleaved caspase 3-positive cells, we predominantly observed the presence of clustered apoptotic cells throughout multiple regions of each Cep120-KO kidney. There are at least two possible explanations for this phenomenon: 1) First, the clusters of cleaved caspase 3-positive cells may be due to enrichment in areas where Cep120-ablated cells (and their derived daughter cells) are present. This makes sense as the loss of

centrosomes would occur during the cell division phase, and the daughter cells would then be nearby. 2) Another explanation of this phenomena might be a study published by Boivin and Bridgewater (AJP-Renal 2018, which we cite in the manuscript) describing clusters of apoptotic cells specifically in the corticomedullary junction of kidneys, following stromal β -catenin deletion. As our stromal deletion of centrosomes leads to deregulation of many of the same signaling pathways (as shown in the Fig. 5G), this could result in the localized cell death patterns that we observed.

Minor point:

In the discussion part, the authors wrote "In a concurrent study". The study has been done in the same lab. Can the authors change the sentence?

- We have adjusted this sentence to indicate that the study was performed by our group.

Dear Moe,

Thank you for submitting your revised manuscript. It has now been seen by all of the original referees.

As you can see, the referees find that the study is significantly improved during revision and recommend publication. However, I need you to address the points below before I can accept the manuscript.

- Please address the remaining minor concern of referee #3 about the missing figure.
- Please rename the "Conflict of Interests" section as "Disclosure Statement and Competing Interests".
- Please remove the Author Contribution section from the manuscript text.
- As per our format requirements, in the reference list, citations should be listed in alphabetical order and then chronologically, with the authors' surnames and initials inverted; where there are more than 10 authors on a paper, 10 will be listed, followed by 'et al.'. Please see <https://www.embopress.org/page/journal/14693178/authorguide#referencesformat>
- We note that the funding information is not complete in the manuscript submission system - UAB-UCSD O'Brien Core Center for Acute Kidney Injury Research (NIH P30-DK079337) has not been entered.
- We note the following about figure callouts: Figure 6F has not been called out. Appendix Figure S1 is called out on p9, but there are not any Appendix figures (as per referee #3).
- Tables EV1-EV4 should be removed from the manuscript together with their legends and should each be uploaded separately as Expanded View Tables. The legends should be included in the excel files.
- Synopsis image needs to be removed from the manuscript file and uploaded separately in jpeg, TIFF or png format. It should be sized 550 pixels wide x 300-600 pixels high.
- Our data editors have asked you to clarify the below points in the figure legends:
 - o Please note that a separate 'Data Information' section is required in the legends of all the figures.
 - o Please note that legend for figure EV1j should be provided on a new line or the legend should be labeled as I-J.
 - o Please note that in figures 1d, f, h; 2e-g; 3b, c, e, f; 4a, d, f, j; 5b-d, f-g; EV1b-c, e-h; EV2d-e; EV3d; EV5a-c, e-f there is a mismatch between the annotated p values in the figure legend and the annotated p values in the figure file that should be corrected.
 - o Please note that the box plots need to be defined in terms of minima, maxima, centre, bounds of box and whiskers, and percentile in the legend of figures 1d, f, h; 2e-h; 3b-c, e-f; 4a, d-f, h, j; 5b-d, f; 6c, d, g, i-k; EV1b-c, e-j; EV 2d-e; EV3b-d, f, g; EV4c-g; EV5a-f.
 - o Please note that the error bars are not defined in the legend of figure 5g.
 - o Please indicate what white arrows represent in the legend of figure EV2c
- Papers published in EMBO Reports include a 'synopsis' and 'bullet points' to further enhance discoverability. Both are displayed on the html version of the paper and are freely accessible to all readers. The synopsis includes a short standfirst summarizing the study in 1 or 2 sentences (max 35 words) that summarize the paper and are provided by the authors and streamlined by the handling editor. I would therefore ask you to include your synopsis blurb and 3-5 bullet points listing the key experimental findings.

Thank you again for giving us to consider your manuscript for EMBO Reports, I look forward to your minor revision.

Kind regards,

Deniz

--

Deniz Senyilmaz Tiebe, PhD
Editor
EMBO Reports

Referee #1:

I would like to express my apologies for the slight delay in my response. Upon thorough review of the revisions and the responses to my comments and to those of the other reviewers, I am pleased to confirm that all the previously raised concerns have been satisfactorily addressed. The additional information and clarifications provided have enhanced the manuscript and have resolved the queries I had.

Referee #2:

I thank the authors for addressing my comments and I particularly appreciate the effort to clarify the experiments shown in

Figure 4, which I still think are difficult to analyse from looking at the tissue sections alone. I have no further issues and consider the manuscript ready for publication.

Referee #3:

In the revised version of their manuscript, Langner et al take into accounts all my comments.

I did not find the file "(Appendix Fig S1)" as mentioned by the authors in the point by point answer (point3).

"We have included single channels of the immunofluorescent images from Fig 4G (separately for p53 and other markers) in the supplementary material "

Nevertheless the manuscript is now suitable for publication

All editorial and formatting issues were resolved by the authors.

Dear Moe,

Thank you for submitting your revised manuscript. I have now looked at everything and all is fine. Therefore, I am very pleased to accept your manuscript for publication in EMBO Reports.

Congratulations on a nice work!

Kind regards,

Deniz

--

Deniz Senyilmaz Tiebe, PhD
Editor
EMBO Reports

--

I am very pleased to accept your manuscript for publication in the next available issue of EMBO reports. Thank you for your contribution to our journal.
